# Repeatability of protein structural evolution following convergent gene fusions

Naoki Konno[1] ✉, Keita Miyake[2], Satoshi Nishino ®[3,4], Kimiho Omae[3,5], Haruaki Yanagisawa[6], Saburo Tsuru ®[7], Yuki Nishimura ®[3], Masahide Kikkawa ®[6], Chikara Furusawa[7,8] & Wataru Iwasaki[1,3]

Convergent evolution of proteins provides insights into repeatability of genetic adaptation. While local convergence of proteins at residue or domain level has been characterized, global structural convergence by inter-domain/molecular interactions remains largely unknown. Here we present structural convergent evolution on fusion enzymes of aldehyde dehydrogenases (ALDHs) and alcohol dehydrogenases (ADHs). We discover BdhE (bifunctional dehydrogenase E), an enzyme clade that emerged independently from the previously known AdhE family through distinct gene fusion events. AdhE and BdhE show shared enzymatic activities and non-overlapping phylogenetic distribution, suggesting common functions in different species. Cryo-electron microscopy reveals BdhEs form donut-like homotetramers, contrasting AdhE's helical homopolymers. Intriguingly, despite distinct quaternary structures and < 30% amino acid sequence identity, both enzymes forms resemble dimeric structure units by ALDH-ADH interactions via convergently elongated loop structures. These findings suggest convergent gene fusions recurrently led to substrate channeling evolution to enhance two-step reaction efficiency. Our study unveils structural convergence at inter-domain/molecular level, expanding our knowledges on patterns behind molecular evolution exploring protein structural universe.

Repeatability is fundamental to science, distinguishing non-random from random phenomena and elucidating causality and predictability[1–3]. Evolution in nature, however, poses challenges for studying repeatability, since we can typically examine historical events that have occurred just once[4,5]. Evolution is characterized by stochastic mechanisms (e.g., mutations and genetic drift), and the adaptation processes are heavily influenced by historical contingencies[4,5]. These

effects decrease the likelihood of repeated evolution, even when subjected to similar selective pressures. Convergent evolution—independent yet similar evolutionary outcomes in distinct lineages—therefore, is a crucial key to discuss evolutionary repeatability.

Through molecular evolution studies, convergence of proteins has been characterized at various levels to decipher driving factors (e.g., selective pressures and evolutionary constraints) and predict-

[1]Department of Biological Sciences, Graduate School of Science, The University of Tokyo, 7-3-1 Hongo, Bunkyo-ku, Tokyo 113-0033, Japan. [2]Department of General Systems Studies, Graduate School of Arts and Sciences, The University of Tokyo, 3-8-1 Komaba, Meguro-ku, Tokyo 153-8902, Japan. [3]Department of Integrated Biosciences, Graduate School of Frontier Sciences, The University of Tokyo, 5-1-5 Kashiwanoha, Kashiwa, Chiba 277-0882, Japan. [4]Atmosphere and Ocean Research Institute, The University of Tokyo, 5-1-5 Kashiwanoha, Kashiwa, Chiba 277-0882, Japan. [5]RIKEN Cluster for Pioneering Research, RIKEN, 2-1 Hirosawa, Wako, Saitama 351-0198, Japan. [6]Department of Cell Biology and Anatomy, Graduate School of Medicine, The University of Tokyo, 7-3-1 Hongo, Bunkyo-ku, Tokyo 113-0033, Japan. [7]Universal Biology Institute, The University of Tokyo, 7-3-1 Hongo, Bunkyo-ku, Tokyo 113-0033, Japan. [8]RIKEN Center for Biosystems Dynamics Research, RIKEN, 6-2-3 Furuedai, Suita, Osaka 565-0874, Japan. ✉e-mail: konno-naoki555@g.ecc.u-tokyo.ac.jp

ability of genetic adaptation[6–8]. At residue level, different lineages of a protein family show independent but same mutations at specific residues under similar selective pressures, such as mutations of ATPα conferring plant toxin resistance to insects[9–11]. At function level, some evolutionarily unrelated enzyme families have evolved similar catalytic activity by acquiring similar active sites, like Ser-His-Asp triad of trypsin and subtilisin[8,12]. Other studies have unveiled structural convergence at domain level (e.g., Cren7/Sul7 and SH3 domains)[13,14]. While all these previous studies revealed convergent evolution at local parts of a protein molecule (i.e. specific residues or domains), convergence of more global three dimensional (3D) structures (i.e., whole structures of protein monomers or multimers) by recurrent evolution of similar inter-domain and inter-molecular interactions remains largely unexplored.

Such protein global structure evolution can be triggered by drastic genetic changes such as gene fusions and fissions[15,16]. These events alter domain architectures of proteins, and especially gene fusion events can further lead to fixation of spatial proximity between previously uncontacted protein domains via linkers[17–19]. The fixed proximity between domains let them collide and contact each other frequently and potentially facilitates global protein structure evolution by acquiring inter-domain interactions. Importantly, gene fusions and fissions can repeatedly occur and same domain architectures can emerge multiple times[20,21]. Although >1% of domain architectures were suggested to have independently emerged twice or more by analyzing thousands of sampled domain architectures[22,23], it is poorly understood whether recurrent 3D structural evolution follows repeated domain architecture emergence.

Alcohol/aldehyde dehydrogenases, comprising aldehyde dehydrogenase (ALDH) and alcohol dehydrogenase (ADH) domains, present suitable targets for studying structural evolution driven by gene fusions. While most eukaryotes, including humans, possess separate genes encoding ALDH and ADH domains, some bacterial species are known to harbor AdhE, a previously known protein family consisting of both ALDH and ADH domains. The ALDH and ADH domains of AdhE are most closely related to EutE and PduQ families, respectively[24,25], and are thought to have emerged through gene fusions of the ancestral single-domain ALDH and ADH genes. With this unique domain composition, AdhE forms helical filament structures as a homopolymer, termed spirosomes initially reported in 1970s[26]. In the recently solved structure of spirosomes, adjacent AdhE molecules exhibit inter-molecular interactions between ALDH and ADH domains[27–30]. This interaction was reported to facilitate substrate channeling between ALDH and ADH domains, passing aldehydes as reaction intermediates[27,28]. This structural arrangement is thereby considered adaptive, because it enables efficient catalysis of the two serial reactions, simultaneously preventing the leakage and diffusion of the cytotoxic aldehydes in cells[27,28]. Furthermore, the pitch of the helical structure has been found to change like a spring triggered by cofactor bindings, proposed as a mechanism of enzyme activity regulation[28,30].

In this study, we report that the bifunctional aldehyde/alcohol dehydrogenase (BdhE) evolved through a gene fusion of ALDH and ADH genes, similarly yet independently to AdhE. Although AdhE and BdhE unshare more than 70% amino acid residues, we demonstrate that BdhE shares functional characteristics with AdhE. We further solve and compare the 3D structures of recombinant AdhE and BdhE, and unveile distinct multimer conformations yet similar inter-domain and inter-molecular interactions between AdhE and BdhE, illuminating the extent of structural convergence after independent gene fusions. We further explore pre-existing genomic contexts of the convergent gene fusions to decipher how similar fusion enzymes independently evolved in bacteria. Our work broadens the current understanding of repeatability in molecular evolution and suggests potentially universal evolutionary patterns for 3D structural evolution of proteins.

## Results

### BdhE is an ALDH-ADH protein family but is evolutionarily distinct from AdhE

Through a sequence homology search for *adhE* across 45,555 bacterial representative genomes from the Genome Taxonomy Database (GTDB)[31], we serendipitously identified a set of genes exhibiting low sequence identity (<30%) but high alignment coverage (>90%) with known *adhE* genes (Fig. 1a). These genes (temporarily named quasi-*adhE*) coded proteins with predicted structures like that of AdhE (Fig. 1b). Notably, the sequence identities were even lower than those partially aligned with AdhEs (45–50% coverage), including single-domain proteins containing only ALDH or ADH (Fig. 1a). Based on these observations, we hypothesized that at least one protein family originated by fusions of ALDH and ADH genes and independently from the fusion event that gave rise to AdhE.

To comprehensively identify proteins with the same domain composition as AdhE, we retrieved all 8467 proteins with ALDH-ADH composition from InterPro database[32]. Notably, only five proteins in InterPro exhibited the reversed ADH-ALDH composition. Ortholog assignment for the ALDH-ADH proteins revealed 44 proteins not annotated as AdhE. Of these, 23 proteins exceeded 800 amino acids in length and exhibited low association with the AdhE family (Supplementary Fig. 1a, b). These findings suggest that these 23 proteins likely belong to Quasi-AdhE families. We used these Quasi-AdhEs as queries to search for homologous genes in 45,555 GTDB genomes (Supplementary Fig. 1c, d). Then, we reconstructed unrooted gene phylogenies for both ALDH and ADH domains of the search hit genes, adding an AdhE sequence from *Escherichia coli* (*E. coli*). Genes with ALDH and ADH domains formed a monophyletic clade apart from the outgroup AdhE in both phylogenies, supported by 100% Ultrafast Bootstrap (UFboot) values (Supplementary Fig. 2). We thereby named this protein family BdhE ("Bifunctional dehydrogenase E").

To elucidate the evolutionary relationship between AdhE and BdhE, we constructed multiple sequence alignments (Supplementary Fig. 3, Supplementary Data 2, Supplementary Data 3) and estimated phylogenetic trees across orthologs with ALDH and/or ADH domain, including AdhEs and BdhEs (Fig. 1c, d). The results revealed that both ALDH and ADH domains of AdhE and BdhE formed distinct clades separated by multiple branches with 99% or more UFboot values. The sister groups of AdhE's ALDH and ADH domains were "acetaldehyde dehydrogenase (acetylating)" (K00132 in KEGG Orthology; including genes annotated as "EutE" consistently with a previous report[24]) and "1-propanol dehydrogenase, *pduQ*" (K13921), supported by 99% and 100% UFboot values, respectively. Although the sister groups of BdhE were not supported by sufficiently large UFboot values (>95%) in Fig. 1c and d, the proteins phylogenetically closest to BdhE were annotated as "aldehyde dehydrogenase" (K00128) and "maleylacetate reductase" (K00217) as sister clades of BdhE with 99% and 100% UFboot values (Supplementary Fig. 2). The non-monophyly of AdhE and BdhE showed that while AdhE and BdhE shared the same domain architecture, they originated from distinct fusion events of evolutionarily distant ALDH and ADH genes. Consequently, AdhE and BdhE present a suitable model case for studying convergent gene fusions and subsequent functional and structural evolution.

### BdhE showed shared enzymatic activity and non-overlapped phylogenetic distribution with AdhE

To elucidate the enzymatic function of BdhE, for which no activity had been experimentally reported or registered in UniProt, we expressed and purified BdhE from *Halomonas eurihalina* and the previously characterized AdhE from *E. coli*[27,28] under the same conditions (Supplementary Fig. 4a). We assayed their activities for ethanol oxidation and acetyl-CoA reduction adjusting to previously reported assay conditions for AdhE[27]. The absorbance at 340 nm, indicative of nicotina-

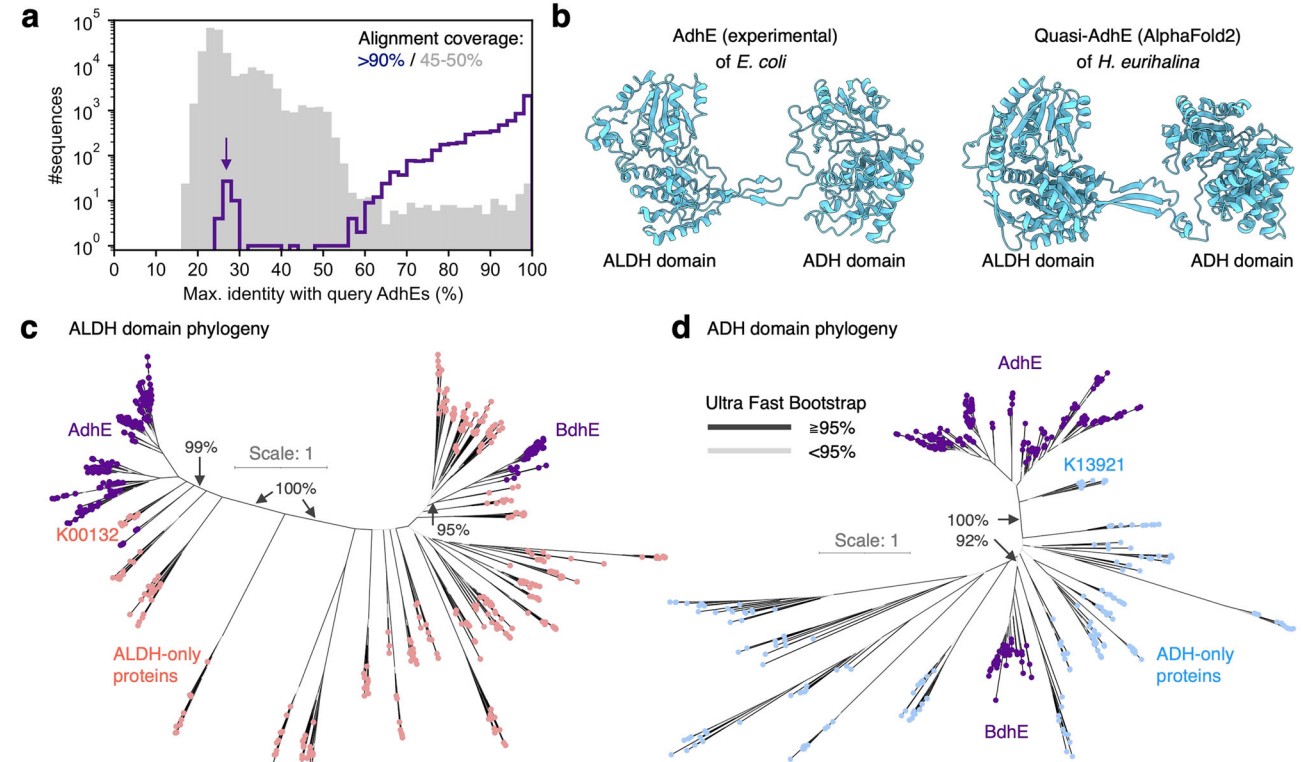

**Fig. 1 | AdhE and BdhE possessing similar yet evolutionarily independent domain architectures. a** Histogram of maximum alignment identities (%) of bacterial proteins hit by sequence similarity search querying known AdhE sequences. The purple line and the grey shade represent the histograms for hit proteins showing >90% and 45–50% alignment coverages, respectively. The arrow indicates the peak of genes showing high alignment coverages but low identities with AdhE (namely Quasi-AdhE). **b** An experimentally determined monomer structure of AdhE (compact form; PDB ID: 6TQM) and a predicted monomer structure of Quasi-AdhE from *Halomonas eurihalina* by AlphaFold v2.3.2. Gene phylogenies of proteins possessing ALDH (**c**) or ADH (**d**) domain. We conducted multiple sequence alignment and phylogeny estimation for the union of 200 randomly sampled AdhEs and all 47 BdhEs (Quasi-AdhEs), as well as 10 proteins randomly selected for every KEGG Ortholog only with ALDH or ADH domains. The red, blue, and purple tips represent ALDH-only, ADH-only, and ALDH-ADH fusion proteins, respectively. The black and grey branches represent splits for which the Ultrafast Bootstrap values were ≥95% or not, respectively. Source data are provided as a Source Data file.

mide adenine dinucleotide (NADH) concentration, changed remarkably upon the addition of either AdhE or BdhE compared to the no enzyme control although BdhE showed lower reaction rates than AdhE (Fig. 2a, b, Supplementary Fig. 4b and c). We also confirmed aldehyde reduction activity of ADH domain (Supplementary Fig. 4b). These results align with reports of reversible ethanol-acetyl-CoA conversion of AdhE[27,28,33]. We further tested choline oxidation activity as BdhE is evolutionarily close to betaine aldehyde dehydrogenase (K00130) (Supplementary Fig. 2, Supplementary Data 1), then found that both enzymes showed the activity (Supplementary Fig. 4c). These results consistently demonstrated that both ALDH and ADH domains in AdhE and BdhE possess shared enzymatic activities. Note that AdhE robustly showed higher reaction rates than BdhE when we varied NaCl concentration across 2–100 mM (Supplementary Fig. 4d).

To explore the functional relationship of AdhE and BdhE in a macroevolutionary context, we compared their phylogenetic distributions across bacteria (Fig. 2c), as functionally coupled/redundant genes are known to show correlated/anti-correlated distribution in diverse species' genomes in general, respectively[34,35]. Both gene families exhibited broad yet sparse distribution across multiple phyla, suggesting the genes have been horizontally transferred across distant clades. Interestingly, none in the analyzed 25,877 species with high-quality complete genomes had both AdhE and BdhE. Furthermore, AdhE and each sister-clade ortholog of BdhE were found to show significantly complementary distribution by EvolCCM[36] ($P < 2.22 \times 10^{-16}$, Fig. 2c, Supplementary Fig. 5a, b), while the distribution of sister-clade families significantly tended to be correlated ($P < 2.22 \times 10^{-16}$), suggesting the two BdhE-sister families are functionally coupled and

involved in redundant functions to those of AdhE. These results consistently suggest that the AdhE and BdhE are functionally redundant, and it is not adaptive to have both.

Given the non-overlapping phylogenetic distributions of *adhE* and *bdhE*, we hypothesized that these genes were distributed in distinct environments. We employed a pipeline to search 16S rRNA gene sequences in metagenomic datasets from diverse environments and scored habitat preferences for each species[37] (Fig. 2d). A comparison of habitat preferences revealed that *bdhE* is enriched in water-soil mixture environments such as activated sludge and marine/freshwater sediments, while *adhE* is enriched in human gut and other human-associated environments (Fig. 2e and Supplementary Fig. 6). These environments are commonly known to be anaerobic or to harbor anaerobic microbes[38–40], suggesting both AdhE and BdhE ecologically contribute to alcohol fermentation in anaerobic conditions, supporting our hypothesis of similar functions in distinct niches. Notably, BdhE was found in saline-associated environments ("marine sediment" and "marine"). Given choline oxidation activity was observed for BdhE, BdhE can also be involved in the synthesis of betaine, a well-known compatible solute for resistance in high-osmotic pressure environments[41–43]. Overall, our results suggest that BdhE can act for similar or redundant functions to AdhE, while it is still possible that each family also have specific ecological roles in different environments.

## BdhE and AdhE evolved similar dimeric structural units forming distinct multimers
As both BdhE and AdhE showed shared enzymatic activities, we further investigated whether BdhE have evolved a similar 3D multimeric

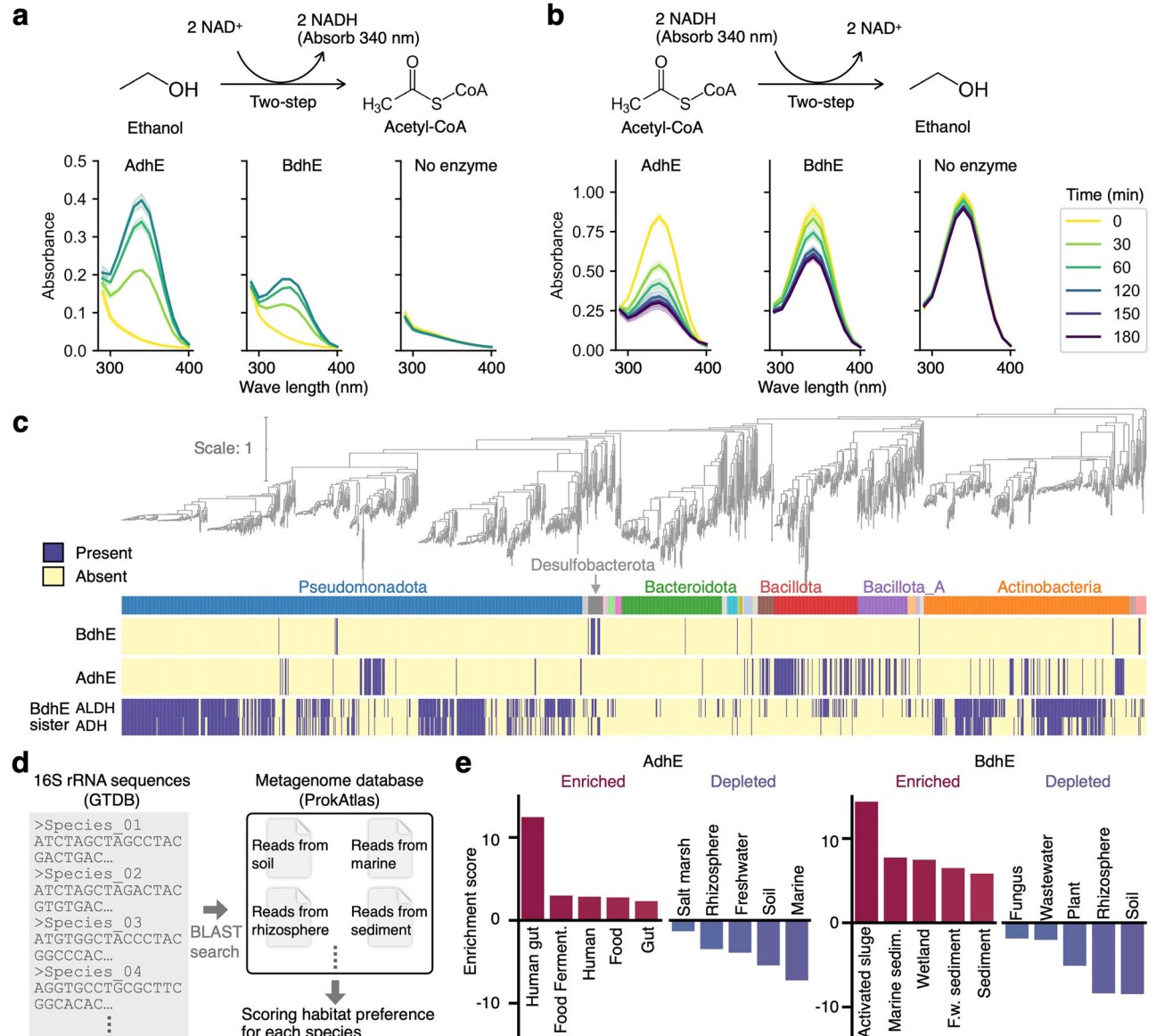

**Fig. 2 | Shared enzymatic activities and non-overlapped phylogenetic and ecological distributions of AdhE and BdhE.** Assay results of ethanol oxidization activity (**a**) and Acetyl-CoA reduction (**b**) for AdhE and BdhE purified by a His-tag affinity column. The progression of the reactions was monitored by NADH-specific absorbance at approximately 340 nm. The yellow-to-purple curves were time-course absorbance spectra for conditions with AdhE, BdhE, or no enzyme. Error bands represent 95% confidence intervals. **c** Phylogenetic distribution of AdhE, BdhE and sister-clade gene families of BdhE. The species phylogenies contain all 43 species with BdhE, all 638 species with BdhE-sisters and 753 species out of 25,877 species with high-quality representative genomes. See Supplementary Fig. 5 for the distribution on full species phylogeny. **d** The schematic diagram of habitat annotation for bacterial species based on 16S rRNA sequences of each species. **e** The habitat enrichment of species with BdhE or AdhE. The enrichment score represents the difference between each environment's average habitat preference scores across species with and without AdhE/BdhE. The habitat preference scores were calculated using the ProkAtlas pipeline[37]. Only top/bottom-five environments are shown. See also Supplementary Fig. 6 for the full results. F.w. and sedim. stands for fresh water and sediment, respectively. Source data are provided as a Source Data file.

structure to that of AdhE following independent gene fusions by biochemical and structural analyses (Supplementary Figs. 7–8). We first verified if both proteins form homocomplex structures by size-exclusion chromatography and blue-native polyacrylamide gel electrophoresis (BN-PAGE). Unexpectedly, BdhE showed a later retention time than AdhE, suggesting that the BdhE complex, if formed, has a smaller molecular size than the AdhE complex (Supplementary Fig. 7a). In addition, a BN-PAGE for BdhE resulted in a single band around the molecular weight of tetramers (391.6 kDa) (Supplementary Fig. 7b). Therefore, BdhE was suggested to form a complex, but it is not polymer like AdhE but a stable tetramer. Notably, the BN-PAGE for AdhE showed bands around the monomer and dimer weights, which is

consistent with previous reports[44], probably because AdhE polymers were trapped at the starting point of the gel and only disassembled molecules were observed as bands.

As BdhE was suggested to form a homotetramer, we solved its quaternary structure using cryo-electron microscopy (cryo-EM), as well as that of AdhE (Fig. 3a, Supplementary Figs. 8, 9a, b, and Supplementary Table 1; gold-standard Fourier shell correlation (GSFSC) resolution of AdhE: 2.87 Å, BdhE: 2.55 Å). The results revealed that BdhE formed a ring-shaped tetramer, while AdhE formed a helical polymer, even though they were purified in the same condition. Two different conformations (extended and compact) have been reported for AdhE, and the structure we solved resembled the compact

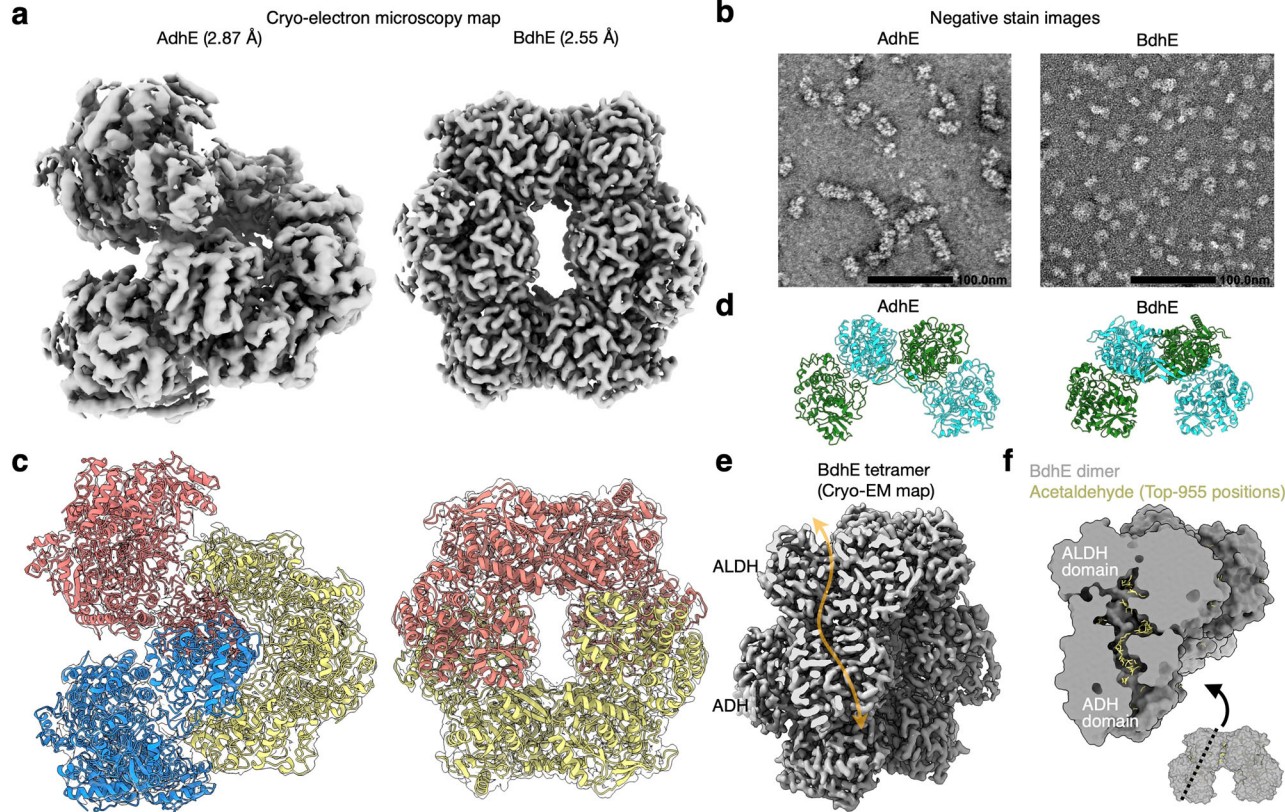

**Fig. 3 | Cryo-electron microscopy structures of AdhE and BdhE suggesting convergence of inter-domain interactions. a** The density maps of AdhE and BdhE. **b** The negative stain electron microscopy analysis of AdhE and BdhE. Scale bars represent 100.0 nm. The images represent only one zoomed-in view of the entire image. See also Supplementary Fig. 7c for the full images. **c** The structural models of AdhE and BdhE superimposed with electron density maps. The red, yellow, and blue parts represent different dimeric structural units. **d** The structure of dimeric structural units of AdhE polymers and BdhE tetramers shown in **c**. Green and cyan molecules represent different protein molecules. **e** A cross-section of BdhE tetramer where ALDH and ADH domains were closely positioned. The transparent orange path represents the tunnel through which substrates pass. **f** Docking simulation results of a BdhE dimer and an acetaldehyde molecule. The yellow molecular structures represent 955 possible docking positions of acetaldehyde within 4 kcal/mol from the lowest-energy position.

conformation, consistent with previous reports that AdhE forms compact conformation without cofactors[28,30]. Negative stain images also supported the structural difference (Fig. 3b and Supplementary Fig. 7c). However, through modeling the structures of BdhE and AdhE, we revealed that both helical and ring-shaped structures consist of similar structural units of dimers (Fig. 3c). Both the BdhE and AdhE dimers showed a bent shape whose bending angles were different (Fig. 3d and Supplementary Fig. 7d). The slight structural difference likely contributes to the substantially distinct multimers, as the larger bending angle in AdhE compared to BdhE may prevent single-turn circle formation and instead promote the assembly of open-ended helical structures. It is also mentionable that AlphaFold[45] accurately predicted AdhE/BdhE homodimer units and even the tetramer structures of BdhE (Supplementary Fig. 7e). However, BdhE's predicted circular structure might have been interpreted as an artifact without cryo-EM results, because AlphaFold preferred predicting ring structures even for AdhE, failing to predict the helical structures.

### BdhE and AdhE convergently evolved channeling interfaces via distinct loop elongation

In the AdhE and BdhE dimers, the proximity between ALDH and ADH domains from different protein molecules was commonly observed, suggesting that BdhE exhibits substrate channeling similar to AdhE by forming the dimeric structure. Looking into the cross-section of the BdhE tetramer's electron density, we found that the tunnels where substrate passes were connected between ALDH and ADH domains

(Fig. 3e). Similarly to a previous study of AdhE[30], we further confirmed that a substrate molecule could be inside the tunnel stably by docking simulation both in AdhE and BdhE dimers (Fig. 3f and Supplementary Fig. 9c), supporting the possibility that both fusion proteins have convergently evolved substrate channeling by independently acquiring ALDH-ADH contacts.

To characterize how AdhE and BdhE evolved the ALDH-ADH contact interfaces, we compared the interaction interface residues in their dimeric structures (Fig. 4a). We found the ALDH-ADH interface was involved with multiple loop structures (loop 1 and 2 of AdhE and loop 3 and 4 of BdhE; Fig. 4b, Supplementary Fig. 9d), which is consistent with a previous report regarding AdhE[28]. Intriguingly, these interface-forming loops of AdhE and BdhE were non-homologous, and the inter-domain interactions were achieved by AdhE- and BdhE-specific manners, respectively. Furthermore, we found that these loop sequences were AdhE- or BdhE-specific and absent even in the sister-clade ALDH or ADH genes of AdhE and BdhE (Fig. 4c), suggesting AdhE and BdhE convergently evolved channeling interfaces using independently elongated distinct loop structures. Therefore, AdhE and BdhE convergently evolved inter-domain interactions at structure level, rather than at residue level.

The difference in channeling interfaces of AdhE and BdhE resulted in structural differences in the tunnel where the substrates would be passed. We found a gap in the tunnel wall formed by a BdhE dimer, which were not found in AdhE dimers (Supplementary Fig. 9e–g). In an AdhE dimer, the linker of ALDH and ADH domains filled the gap, while

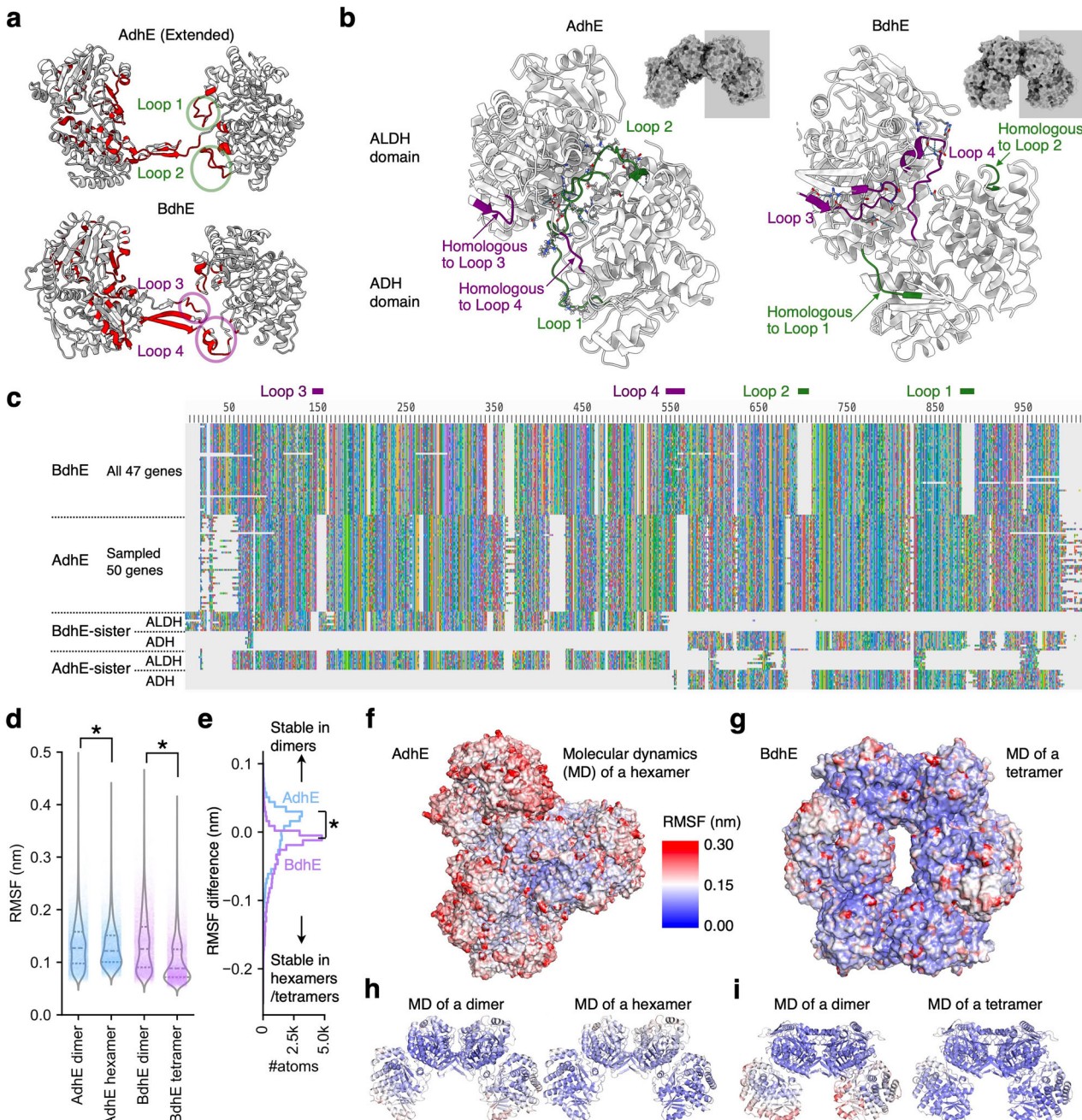

**Fig. 4 | Distinct properties of convergently evolved AdhE and BdhE multimers.**
**a** The residues involved in inter-molecule interactions of AdhE and BdhE dimers.
The red residues represent the interaction interfaces estimated by PICKLUSTER
(without-clustering mode) implemented in ChimeraX-v1.7rc202311290355[76]. The
AdhE dimer structure of the extended conformation was retrieved from the Protein
Data Bank (PDB ID: 6TQH). Loops 1-4 were annotated to the particular loop
structures where inter-molecule interaction residues of AdhE or BdhE enriched.
**b** The ALDH-ADH interaction interface of AdhE and BdhE. The green and purple
loop structures represent the loops 1-4 and their homologous parts. **c** Multiple
sequence alignment of BdhE, AdhE, and sister-clade single-domain families of them.
All 47 BdhEs, randomly sampled 50 AdhEs, and randomly sampled 10 genes for
each of the other families are shown. The positions corresponding to loops 1-4 are
indicated by green and purple bars. **d** Comparison of RMSF distributions among
dimers and hexamer/tetramer of AdhE and BdhE, respectively. RMSFs were

calculated for every atom in each replicate run, and averaged across replicate runs.
The RMSFs of an AdhE hexamer were calculated for the intermediate dimer unit.
The asterisk indicates the statistical difference (Two-sided Wilcoxon signed rank
test. $P = 0.0035$ and $<2.23 \times 10^{-308}$ for AdhE and BdhE, respectively.). **e** Histograms
of RMSF differences for each atom between the AdhE hexamer and dimer, and
between the BdhE tetramer and dimer. Positive and negative values indicate atoms
that are more stable (i.e., have lower RMSF) in the dimer and in the hexamer/
tetramer, respectively. The asterisk indicates the statistical difference (Two-sided
Mann-Whitney U test. $P = <2.23 \times 10^{-308}$.). Color mapping of RMSF onto the 3D
structures of an AdhE hexamer (**f**), a BdhE tetramer (**g**), an AdhE dimer (**h**), and a
BdhE dimer (**i**). In Fig. 4h and i, RMSFs calculated by molecular dynamics (MD) for
dimer structures are shown on the left, and RMSFs for hexamer/tetramer structures
mapped onto dimer structures are shown on the right, respectively. Source data are
provided as a Source Data file.

the linker of BdhE formed the elongated loop (loop 2 in Fig. 4a and b) and thereby remained the gap. In BdhE, however, another BdhE dimer partially filled the gap in the tunnel wall within the tetrameric complex (Supplementary Fig. 9f), suggesting that the tetrameric structure is more effective at preventing intermediate aldehydes from leaking out than the dimeric structure.

## The difference between helical and circular multimer structures is associated with structural stability

Because the dimeric structural units of AdhE and BdhE form distinct multimeric assemblies−namely, a helical polymer in the case of AdhE and a circular tetramer in BdhE−we hypothesized that these overall structural differences would lead to distinct characteristics in structural stability. To test this, we conducted 20 ns molecular dynamics (MD) simulations of the AdhE hexamer, BdhE tetramer, and their corresponding dimers (Supplementary Movie 1, Supplementary Fig. 10a). To verify whether the systems had reached a stable state, we confirmed that the distributions of RMSD changes per 10 ps interval during the 15−20 ns period approximated normal distributions centered around zero (Supplementary Fig. 10b, c). To assess structural stability, root mean square fluctuations (RMSFs) were calculated for each atom over the same time interval (Fig. 4d). For the AdhE hexamer, RMSFs were calculated for the dimer that was not located at either end of the helix. The results showed that the both AdhE hexamer and BdhE tetramer showed significantly decreased RMSF, or increased the structural stability, compared to AdhE and BdhE dimer, respectively (Fig. 4d). Importantly, the RMSF reductions were significantly larger for BdhE tetramerization than AdhE hexamerization (Fig. 4e). By mapping the RMSF values onto the 3D structures of AdhE and BdhE, we found the BdhE showed higher structural stability than the AdhE across the whole structure (Fig. 4f and g). We further found that the stabilization by forming tetramers of BdhE was observed for most parts of the structures, while that by forming hexamers of AdhE was observed only for ADH domains (Fig. 4h and i). These results indicate that the tetramer and polymer formation by BdhE and AdhE, respectively, can stabilize the structures compared to forming dimers, but the stabilization effect is larger for BdhE, possibly due to the flexibility difference between circular and helical structures.

## Genomic neighboring of ALDH and ADH genes underlies independent gene fusions

As distinct pairs of ALDH and ADH genes fused twice into *adhE* and *bdhE*, we next asked whether any common genomic background facilitated fusions of those genes. Gene fusions often occur between genes neighboring in genomes[15], so we hypothesized that single-domain ALDH and ADH genes tend to be neighbors in bacterial genomes. We comprehensively detected neighboring gene pairs located within ten genes for various ALDH and ADH orthologs across 26,778 high-quality representative genomes of prokaryotes and found that neighboring genes are enriched in specific ALDH and ADH ortholog pairs (Fig. 5a). The adjacent gene pairs include *pduP*-*pduQ* (KEGG ID: K13922-K13921), *eutE*-*eutG* (K04021-K04022), and *gbsA*-*gbsB* (K00130-K11440), which have been reported to be coded in the same operons and to be functionally coupled[42,46,47]. These results suggest that functionally coupled ALDH and ADH genes repeatedly become neighbors in genomes and are transcribed together in the same mRNA to adjust the expression levels of enzymes for a series of metabolic reactions.

Based on phylogenies of ALDH and ADH domains (Fig. 1c and d, Supplementary Fig. 2), we have identified pairs of orthologs, K00132-K13921 and K00128-K00217 as ALDH- and ADH-only sister clades of AdhE and BdhE, respectively. By analyzing the neighboring gene pairs, we noticed that the sister-clade families of AdhE/BdhE tend to be coded by neighbor genes, respectively (Fig. 5a). The synteny around these genes were conserved and they were adjacent to genes with potential functional relationships with ALDH/ADH genes (Fig. 5b). For

example, the pairs of AdhE sisters were close to *eutM* coding major elements of bacterial microcompartments where ethanol metabolism is catalysed[48], and BdhE sisters were also close to *gctA/B* (gluconate CoA transferase) which may contribute to produce carbonyl-CoA as substrates of an ALDH. The result suggests the ancestral single-domain genes of both *adhE* and *bdhE* were originally coded in genomically close positions before gene fusion and thereby enabled to fuse by small deletions. Although the functions and expressions of the sister-clade families were seemingly coupled, the adjacently coded proteins were not predicted to show inter-domain interactions, and the results were robust for any other pair of ALDH and ADH families (Fig. 5c, Supplementary Fig. 11a). In contrast, split domains of AdhE and BdhE as positive controls were predicted to interact with each other (Fig. 5d). Therefore, the convergent ALDH-ADH interactions observed in AdhE and BdhE were suggested to have not evolved before gene fusions through acquisitions of the family-specific loops (Fig. 4a). Notably, the predicted interaction of BdhE cannot be an artifact biased by AdhE complex structures learned by AlphaFold, because the interaction interface of BdhE is non-homologous to that of AdhE (Fig. 4b), and because the heteromeric interaction was not predicted for sister-clade families of AdhE, which are evolutionarily closer to AdhE than BdhE (Fig. 5c). Regarding homomeric interactions, all the single-domain proteins were predicted to form homodimers whose interaction interfaces resembled ALDH-ALDH and ADH-ADH interfaces in both BdhE and AdhE (Supplementary Fig. 11b, c, and 12a, b), suggesting the helical or circular multimers of AdhE and BdhE were achieved by homomeric interactions inherited from ancestors and heteromeric interactions convergently acquired.

## Discussion

We identified BdhE, an ALDH-ADH protein family that emerged by gene fusion independently of AdhE. We further showed that AdhE and BdhE convergently evolved ALDH-ADH interactions with distinct loop elongations while their multimer structures highly diverged (a helical polymer and a circular tetramer). Based on these findings, we propose a model of AdhE/BdhE evolution (Fig. 5e), in which the repeated evolution of AdhE and BdhE would have started from distantly positioned genes coding ALDH and ADH enzymes which form homodimers and/ or homotetramers[49] (Supplementary Fig. 11b, c). Then, some pairs of the genes became neighbors in genomes (Fig. 5a), and the neighbor genes were fused (Fig. 1b−d). The fused protein would show no heteromeric interactions at first, like their closest relative single-domain families (Fig. 5c). Then, the fused enzyme families convergently evolved elongated loops and interactions between ALDH and ADH domains, suggested to have enabled substrate channeling (Fig. 3e and f). As a result of the inter-domain interaction, AdhE evolved spring-like helical polymers with variable pitch lengths while BdhE evolved ring-like tetramers, both using the interaction interfaces for newly evolved ALDH-ADH contacts and ancestral homomer formation (Fig. 3a−c). In this way, convergent gene fusions led to convergence of inter-domain contacts and dimeric structures while divergence of overall multimer structures.

This stepwise repeated evolution of AdhE and BdhE suggests evolutionary patterns of protein structures and underlying their principles. Unlike well-documented functional convergence between distinct protein families and residue-level convergence within families[6,8], our study provides a model case of structural convergence through the evolution of inter-domain and inter-molecular interactions.

Our findings suggest three evolutionary patterns. First, gene neighboring precedes fusion protein evolution. This pattern can be common across many different fusion gene families[15], as the neighboring genes can be co-regulated and horizontally transferred together within operons. In the case of ALDH and ADH, the coordination of gene expression levels can be adaptive by preventing the accumulation of cytotoxic aldehyde as intermediates. Proximities of genes in

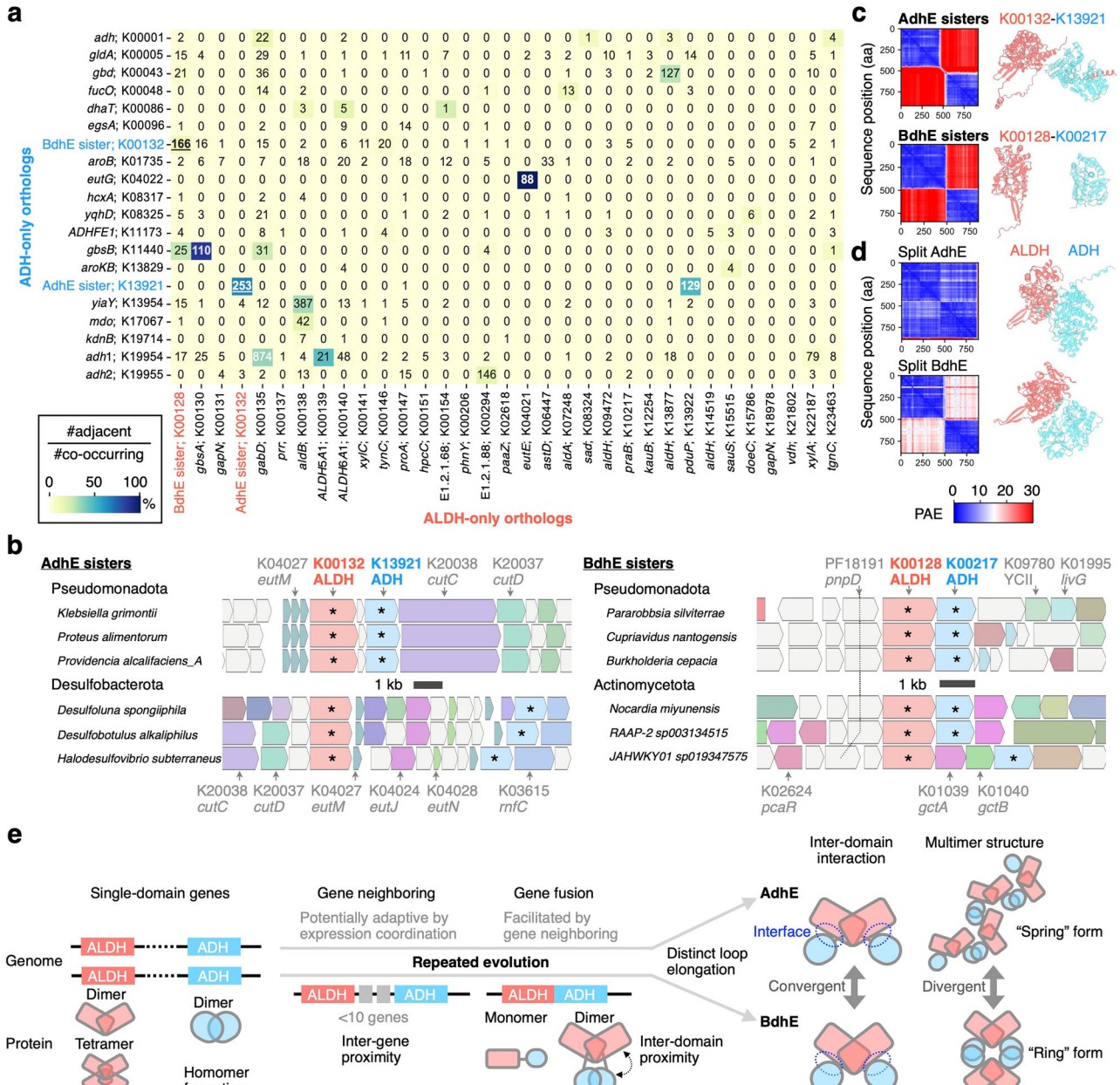

**Fig. 5 | Genomic evolution preceding recurrent gene fusion events and the model for convergent evolution of AdhE and BdhE. a** A heatmap showing the counts/ratios of the pairs of ALDH and ADH genes adjacently coded in prokaryotic genomes, i.e., coded in the same strand of the same contigs intervened by <10 genes. The annotated numbers indicate the gene pair counts in 26,778 prokaryotic genomes, and the color scale represents the ratio of the number of adjacent gene pairs per total number of pairs co-existing in the same genomes. The ALDH and ADH orthologs in this panel correspond to ALDH/ADH orthologs included in the phylogeny of Fig. 1c and d. The number of gene pairs of AdhE- and BdhE-sister orthologs are underlined. **b** Multi-species comparison of gene synteny around AdhE- and BdhE-sister orthologs. Here we sampled three species in different genera for two phyla, whose genomes have AdhE- and BdhE-sister orthologs as neighbors (genes with asterisks). The colored genes are assigned KEGG ortholog identifiers. The visualization was conducted by Annoview version 2.0[89]. **c** ColabFold prediction of heteromeric interaction between AdhE-sister or BdhE-sister ALDH and ADH proteins adjacently coded in genomes. Structural predictions were conducted by ColabFold v1.5.2. Here, we show the results of an example protein pair for the AdhE and BdhE sisters, respectively. The left heatmap and the right structures indicate the positioned aligned errors (PAEs) and the predicted structures where ALDH and ADH proteins are colored in red and cyan, respectively. **d** ColabFold structure prediction of heteromeric interaction between split AdhE/BdhE domains. Here, we show the results for AdhE and BdhE used in our enzymatic and structural analysis. **e** A schematic model of repeated genomic and structural evolution leading to extant AdhE and BdhE.

genomes can also facilitate gene fusions by reducing the fitness loss of deletion achieving the fusions.

As the second pattern, gene fusions precede structural convergence of inter-domain interactions. Gene fusion can fix the proximity of previously uncontacted domains and stabilize incomplete inter-domain interactions to facilitate the following inter-domain contact evolution. While a previous study proposed domain

interaction evolution precedes gene fusions[50], the case of AdhE and BdhE suggests the opposite order is also possible. Note that AdhE and BdhE showed similar orientations of interacting ALDH and ADH domains, while substrate channeling could be achieved by other orientations (i.e., rotating a domain keeping the substrate channel), which may suggest the particular orientation of domains were favored during evolution because of any evolutionary constraints such as

thermostability of the protein structures[51]. As the third pattern, structural innovation during evolution such as inter-domain interactions are achieved by an elongated loop, as reported in the evolution of enzymatic activities, too[52]. Existing structures constrain evolutionary changes, so that evolution tend to favor alterations in loop structures that don't disrupt existing structural elements.

These findings and implications regarding structural convergence following independent gene fusions may be generalizable to other protein families. Indeed, previous studies have shown >1% of analyzed domain architectures could be candidates of convergent emergence[22,23]. Given >20,000 annotated protein domains (Pfam v37.3[53]) and numerous potential fusions among them, there are many candidates of convergently evolved fusion proteins, in which we may observe inter-domain interactions converged following repeated gene fusions. Indeed, a recent study showed convergent domain fusions of molybdenum insertase and the AlphaFold prediction supported proximal positions of the two domains[54]. If these evolutionary phenomena commonly occur across diverse protein families, the three explainable evolutionary patterns described above may provide clues for predicting the future evolution of existing proteins—such as which proteins are likely to undergo fusion and what structural changes may follow these gene fusions.

Our study also shed light on bacterial adaptation strategies to mitigate aldehyde toxicity, which is universal for diverse organisms including humans[55,56]. The results suggest neighboring of ALDH and ADH genes in genomes and ALDH-ADH channeling to prevent aldehyde leakage from enzymes as versatile aldehyde mitigation mechanisms in bacteria. Some of the neighbor ALDH and ADH genes code enzymes known to be confined in metabolosomes, which are also thought to act as aldehyde mitigation[48,57], suggesting multiple distinct mechanisms have been evolved to avoid aldehyde accumulation in bacterial metabolic evolution. Substrate channeling within molecules of AdhE or BdhE can be a more cost-effective strategy than producing the massive apparatus of metabolosomes.

While the ALDH-ADH interaction convergently evolved, AdhE and BdhE showed distinct quaternary structures (i.e., "spring" and "ring" form). The finding suggests that the whole multimeric structures can substantially diverge by slight difference of structural units, such as the bending angle of dimeric units. The spring-ring differences between the two families potentially result in the functional properties of the enzymes. While spring-form AdhE has been suggested to regulate its activity depending on cofactor binding by dynamically changing the helix pitch lengths[28,30], BdhE forms a ring structure which will not show such structural freedom observed in helical structures (i.e. dynamic pitch length) (Fig. 4d–i). The ring structure of BdhE is thus suggested to confer more rigid enzymatic activity than AdhE against environmental changes (e.g., salinity, pH, or temperature). Therefore, the spring-ring difference can be caused by distinct selective pressures for flexibility or stability of enzymatic functions, although it could also be a result of neutral evolution. Indeed, species with BdhE inhabit diverse natural environments, including marine, freshwater, hydrothermal vents, and hot springs, although those with AdhE mainly inhabit human-associated stable environments (Fig. 2e and Supplementary Fig. 6). As proteins showing rigid structures are generally found in thermophiles[58], the multimeric structural difference between AdhE and BdhE can be associated with ecological niche difference. In addition, the lower enzymatic activities of BdhE compared to those of AdhE (Supplementary Fig. 4b, c) may reflect a general stability-activity trade-off observed across diverse enzymes[59,60]. Notably, the structural stability of AdhE and BdhE complex structures were evaluated based on molecular dynamics simulation conducted under a moderate temperature condition (300 K). It would also be beneficial to experimentally assess thermostability by thermal shift assays[61], temperature-dependent measurements of isothermal titration calorimetry[62], and/or dynamic light scattering measurements[63].

Several protein families, including Rad51[64], MuB[65], CK2[66], and allophycocyanin[67], exhibit both helical (or linear) polymer and circular multimer structures, with functional differences reported between these conformations. The helical-circular transitions in allophycocyanin, analogous to the AdhE-BdhE structural divergence, stem from variable bending angles of structural units, suggesting a general molecular mechanism leading to multimeric structure diversity across protein families. Rad51, MuB, and CK2 were shown to have conformational plasticity within individual molecules, and allophycocyanin transitioned between stable conformations during evolution within the protein family[64–67]. The evolution of AdhE and BdhE, on the other hand, represents a unique case of helical-circular divergence by convergent evolution of inter-domain interactions in independently emerged fusion proteins. This convergent evolution revealed in this study offers valuable insights for exploring alternative amino acid sequence spaces that encode similar inter-domain interactions, hinting the design of proteins with comparable monomer structures but orthogonal multimeric assembly into ring and spring conformations. It is also noteworthy that AdhE and BdhE may have undergone transitions between different assembly states (helical and circular conformations) within each family, considering recent evidences showing protein families experience changes in their quaternary structure during evolution[67,68].

In this study, we demonstrated a clear example of macroscopic protein structure convergence, where similar dimeric structures evolved through independently acquired interactions. This finding extends our understanding of protein convergence beyond residue and domain levels to inter-domain and inter-molecular levels. Further investigation of protein structural evolution, particularly in proteins with convergent domain architectures, may uncover hidden evolutionary repeatability of inter-domain interactions. The latest rapid expansion of protein structure universe—driven by advances in cryo-electron microscopy technologies and structure prediction tools like AlphaFold and ESMFold[45,69,70]—has opened unprecedented era for elucidating the patterns and principles underlying protein structural evolution.

## Methods

### Dataset

We retrieved all the protein sequences of every prokaryotic representative genome and reference phylogenies from GTDB r202 on April 28, 2021. The datasets contained 45,555 bacterial and 2339 archaeal species of all the phyla defined in GTDB. We annotated orthologs for each gene in all the representative genome based on KEGG Orthology by KofamScan v1.3.0[71]. We also extracted a tree of all the 25,877 bacterial species for which high-quality representative genomes (defined as >95% completeness and <5% contamination throughout this study) are available in GTDB, using the TreeNode.prune function ("preserve_branch_length = True") in ete3 toolkit 3.1.2 (51). 16S rRNA gene sequences for 22,304 of the 25,877 species were also downloaded from GTDB r202.

### Extraction of non-AdhE ALDH-ADH fusion proteins

As described previously[72], we conducted sensitive sequence similarity search of AdhE against all the proteins coded in 45,555 bacterial genomes from GTDB using MMseqs v13.45111 (easy-search -s 7.50), by querying 4833 proteins annotated as AdhE (K04072) by KofamScan and showing >800 aa length in GTDB genomes. Because the sequence similarity search suggested existence of non-AdhE ALDH-ADH fusion proteins, we next retrieved all the 8472 protein sequences with ALDH (PF00171) and ADH domain (PF00465) from InterPro (retrieval date: November 29, 2022), in which 8467 and 5 proteins had ALDH-ADH and ADH-ALDH domain architecture, respectively. After extracting ALDH-ADH proteins which include AdhE, we annotated KEGG Orthology identifiers by KofamScan v1.3.0. Then, we extracted 23 proteins not

annotated as AdhE (K04072 in KEGG Orthology) and with >800 aa length as non-AdhE ALDH-ADH proteins. We next searched their homologs in bacteria by querying N-terminal 480 aa (corresponding to ALDH domains) or C-terminal 370 aa (corresponding to ADH domains) of those 23 proteins against all the proteins coded in 45,555 bacterial genomes from GTDB. The sequence searches resulted in 7,370,016 and 1,718,637 proteins hit by querying the N-terminal and C-terminal sequences, respectively. From the hits, we extracted non-AdhE ALDH-ADH proteins which showed >40% sequence identity for both N-terminal and C-terminal queries. We also obtained single-domain close homologs of the non-AdhE ALDH-ADH proteins by extracting hits showing >50% and >40% alignment identity for N-terminal and C-terminal queries, respectively. Lastly, we aligned those ALDH-ADH proteins and their close homologs by MAFFT v7.310 (no option), and added an AdhE sequence (from the *E. coli* genome in GTDB) to the alignment as an outgroup by MAFFT (--add). We next trimmed alignment columns dominated by gaps with TrimAl v1.4rev15 (-gappyout). Then, we constructed gene phylogenies for both protein domains by IQTree v2.0.3 (-m MFP -bb 1000 -nt 20). We extracted 47 proteins showing >700 aa as BdhE and confirmed their monomer structures predicted by AlphaFold are all similar to each other excluding two proteins with unidentified amino acids. We named the extracted ALDH-ADH fusion proteins as BdhE (Bifunctional dehydrogenase E).

## Phylogenetic analysis of AdhE and BdhE across diverse ALDH and ADH proteins

To unveil the phylogenetic relationship between AdhE and BdhE among diverse ALDH and ADH protein families, we firstly extracted KEGG Orthologs annotated for genes coding ALDH (Pfam ID: PF00171) or ADH domains (Pfam ID: PF00465 ("Fe-ADH") or PF13685 ("Fe-ADH_2")) in KEGG Genes database (retrieval date: March 31, 2022). We excluded 13 of the extracted orthologs (K00254, K00891, K01999, K01588, K01647, K03106, K03110, K03469, K03601, K07029, K13821, K13829, and K13830 in KEGG Orthology), because they coded non-enzymatic genes and/or false positives caused by genes fused with other ALDH or ADH genes. The ALDH- and ADH-only KEGG orthologs after the curation are shown in Fig. 5a. Next, we collected protein sequences of randomly selected 10 genes in GTDB bacterial genomes for each of the KEGG Orthologs. Then, we merged the protein sequences of single-domain orthologs, randomly sampled 200 AdhEs, and all the 47 BdhEs, and reconstruct the domain-wise gene phylogenies. Multiple sequence alignment, trimming, and phylogeny estimation were done by MAFFT v7.310 (--thread 20), Trimal v1.4rev15 (-gt 0.7), and IQTree v2.0.3 (-m MFP -bb 1000 -nt 10). To test if AdhE and each of BdhE sister families (K00128 and K00217) significantly tend to show correlated or anti-correlated distribution in bacterial genomes, we used EvolCCM v0.1.0[36] to construct a model for gains and losses of AdhE, K00128, and K00217 across the reference genome phylogeny.

## Habitat preference analysis of whole bacterial species

To investigate environments where species with AdhE or BdhE enriched in, we estimated the habitat preference for each of the 22,304 bacterial species for which high-quality representative genomes and 16S rRNA sequences were available from GTDB (see Datasets). We queried the 16S rRNA sequences of the 22,304 species against ProkAtlas online[37] on June 30, 2021. ProkAtlas evaluates habitat preferences by comparing 16S rRNA sequences to short-read metagenomic data from various environments. Based on these results, ProkAtlas assigns a habitat preference score for each environment for every species. We used BLAST searches with specific criteria: 97% nucleotide identity and a minimum of 150 bp coverage.

## Expression and purification of AdhE and BdhE

The full-length *adhE* gene from *Escherichia coli* (*E. coli*) K12 strain (UniProt ID: P0A9Q7) and the full-length *bdhE* gene from the

*Halomonas eurihalina* genome in GTDB were artificially synthesized with codons optimized for expression in *E. coli*. Each gene was fused with an N-terminal 6-His tag followed by a thrombin cleavage site (LVPR | GS) and inserted into the pET28a (+) vector (Twist BioScience)). Both AdhE and BdhE were expressed in BL21 (DE3) cells by induction with 1 mM isopropyl β-D-thiogalactopyranoside (IPTG) at 18 °C overnight after the preculture until the OD reached 0.4–0.5 at 37 °C in LB medium. Then, we harvested cells by centrifugation at 5000 rpm 4 °C for 15 min, resuspended and sonicated in a lysis buffer (buffer A) containing 50 mM Tris-HCl pH 8.0, 500 mM NaCl, and 5% (v/v) glycerol with 2 mg/mL lysozyme. We clarified the sonicated lysate by centrifugation at 15,000 g, 10 min and 4 °C and filtration with 0.2 μm filter. Then, we conducted His-tag purification using TALON Spin Columns purification (Takara, Cat. No. 635603). We washed the TALON beads with HisTALON Equilibration buffer or Buffer A, and loaded the filtered lysate after sonication. Then, we washed the beads with buffer A containing 50 mM imidazole. The bound AdhE or BdhE proteins were eluted by buffer A containing 100, 150, and 200 mM imidazole. We removed imidazole from the eluates by dialysis in 100 mM NaCl, 1 mM DTT, and 0.5 mM EDTA at 4 °C overnight. We further purified AdhE and BdhE by gel-filtration chromatography column (Cytiva; 28-9909-44; Superdex 200 Increase 10/300 GL) equilibrated with buffer containing 50 mM Tris-HCl pH 8.0 and 100 mM NaCl. AdhE- or BdhE-containing fractions were collected and further concentrated with Amicon Ultra30K MWCO centrifugal filters (Millipore; UFC903024) up to >4 mg/mL. The purified samples were flash-frozen for storage in liquid nitrogen before storage at −80 °C. For proteins used for cryo-electron microscopy (cryo-EM) experiment, we did not freeze the purified protein samples until the cryo-EM sample preparation.

## Enzymatic activity assay for AdhE and BdhE

To investigate the enzymatic activities of AdhE and BdhE, we quantitated the decrease or increase of absorbance by nicotinamide adenine dinucleotide (NADH) peaked at a wavelength of ~340 nm, referring to previous protocols for AdhE enzymatic assay[27]. We used a plate reader (Tecan; Infinite 200 Pro, M Nano) to measure absorbance spectra across 230–400 nm with two replicates (Fig. 2a, b), and used another plate reader (Tecan; Infinite F200 Pro) to measure absorbance values at $360 \pm 35$ nm with four replicates (Supplementary Fig. 4b-d). All assays were conducted at 37 °C, and the total volume was 100 μL (Fig. 2a, b) or 40 μL (Supplementary Fig. 4b-d). In Fig. 2a, b, we measured the activities of the AdhE and BdhE for the forward reaction (oxidization of NADH) in a mixture containing 50 mM Tris-HCl pH 8.0, 50 μM FeSO₄, 200 μM acetyl-CoA, 9–11 mM NaCl, and 500 μM NADH with 6 μg of AdhE or BdhE purified by the histidine tag. For the reversed reaction (reduction of NAD⁺), we assayed the activity in a mixture containing 50 mM Tris-HCl pH 8.0, 50 μM FeSO₄, 200 μM CoA-SH, 20 mM NaCl, 500 μM NAD⁺, and 200 mM ethanol with 22 μg of AdhE or BdhE. In Supplementary Fig. 4b–d, while most conditions were the same as Fig. 2a and b, the concentration of substrate (acetaldehyde or acetyl-CoA) was 10 mM for forward reaction, and the amount of AdhE or BdhE was 1.8 μg (forward reaction) or 2.0 μg (reverse reaction), which was purified by the histidine tag and the gel-filtration chromatography. In addition, Supplementary Fig. 4b and c, we set NaCl concentration as 5 mM for the forward reaction, and 10 mM for the reversed reactions. In Supplementary Fig. 4d, we varied NaCl concentration across 2, 10, 100 mM for ethanol oxidation assays.

## Sodium dodecyl sulfate- and blue native- polyacrylamide gel electrophoresis (SDS-PAGE and BN-PAGE)

We conducted SDS-PAGE using a 12.5% polyacrylamide gel at 21.0 mA for ~45 minutes (ATTO; WSE-1025). Products and intermediates of His-tag purification of AdhE and BdhE were loaded after adding loading buffer (ATTO; AE-1430) and heating at 100 °C for 5 minutes. The molecular mass of each band was estimated based on molecular

weight markers (Broad; 3452). We also conducted BN-PAGE using 3–14% polyacrylamide gradient (ATTO; UH-T314) at 150 V (constant V) for ~75 minutes in a running buffer (ATTO; WSE-7057). Purified AdhE 5 or 10 μg and BdhE 1 or 2 μg were applied with loading buffer (ATTO; WSE-7011). The molecular mass of the AdhE/BdhE multimers was estimated based on molecular weight markers (ATTO; WSE-7016). A BN gel lane was destained in a water solution containing 50% methanol and 12.5% acetic acid, and then re-stained with EzStain Aqua (ATTO; AE-1340).

### Negative stain experiments
To prepare grids for negative stain observation, we applied 3 μL of 50 ng/μL purified AdhE or BdhE to an elastic carbon-coated Cu (250 mesh) grid (Okenshoji; ELS-C10), and dried the grid with filter paper. Then, we stained the grid with 3 μL of 1% uranium acetate three times and dried the grid at room temperature. Excess uranyl acetate was eliminated by filter paper. Then, the prepared grids were observed using a JEM-1400Flash electron microscopy with sCMOS camera (Matataki Flash).

### Cryo-electron microscopy experiments, single particle analyses, and model constructions
To prepare grids for cryo-electron microscopy, we applied 3 μL of 2.02 mg/mL AdhE or 1.35 mg/mL BdhE to a Quantifoil holey carbon grid (R1.2/1.3, 300 mesh copper, Quantifoil Micro Tools GmbH), which was hydrophilized by PIB-10 plasma ion bombarder (Vacuum Device) in advance. Using Vitrobot Mark IV (ThermoFisher Scientific), we blotted the protein samples with a setting of 10 blot force at 6 °C in 100% humidity and froze them immediately in liquid ethane, setting "wait time" and "blot time" as 3 and 4 seconds respectively. Then we collected micrograph images in movie mode using Talos Arctica G2 (ThermoFisher Scientific) with a K2 direct detector (Gatan) operated at 200 kV, 1.03 Å/pixel, 50 e⁻/Å²/micrograph with a −1.75 to −1.0 μm defocus range and 50 movie frames. As shown in Fig. S7C, we conducted single particle analysis for AdhE and BdhE by CryoSPARC v4.4.1[73]. Note that CTF refinement was performed simultaneously with all the homogeneous refinements in CryoSPARC. Molecular structure models were built with Coot v0.9.8.91[74] started from a previously reported model (PDB ID: 6TQM) and an AlphaFold-predicted model of AdhE and BdhE, respectively. We then refined with the real-space refinement procedure implemented in Phenix v1.20.1-4487[75]. Electron density maps and molecular structure models were visualized by UCSF ChimeraX v1.7rc202311290355[76]. For validation of the models, we used MolProbity v.4.5.2[77] to calculate the MolProbity score and Clash score, and used Phenix v1.20.1-4487[75] to calculate correlation coefficient (CC) mask, box, peaks and volume values.

### Docking simulation
To conduct docking simulation, we prepared dimeric structure models of the extended and compact form of AdhE from the Protein Data Bank (PDB ID: 6TQH and 6TQM) and the dimeric BdhE model determined by this study. The models were firstly aligned by Matchmaker algorithm implemented in ChimeraX v1.7rc202311290355[76]. The proteins and acetaldehyde structural models for docking simulation were prepared using Openbabel v3.1.0[78] and UCSF Chimera v1.17.3. The docking simulation was performed using AutoDock Vina v1.2.5[79,80] with a box size of X = 140.0 Å, Y = 100.0 Å, and Z = 100.0 Å. The parameters of exhaustiveness, number of modes, and energy range were set to 100, 1000, and 4 kcal/mol, respectively.

### Molecular dynamics simulations
All molecular dynamics simulations were performed using the GRO-MACS (version 2023.2) molecular dynamics simulation package[81]. The initial atomic coordinates of AdhE hexamer (compact form) were obtained from the PDB (PDB ID: 6AHC), and the model of BdhE was

constructed in this study. These structures were converted into topology and coordinate files utilizing GROMACS. The CHARMM36m force field was used for the protein[82], and the CHARMM General Force Field was used for the ligand[83]. Each system was placed in a dodeca-hedral simulation box, ensuring that all protein atoms were at least 1.0 nm from the box edges, and the box was then filled with TIP3P water molecules[84]. Sodium ions were added to neutralize each system. Energy minimization was performed using the Steepest Descent algorithm in GROMACS. The LINCS algorithm was applied to constrain all bond lengths involving hydrogen atoms[85]. Particle Mesh Ewald was employed to calculate the long-range electrostatic interactions. The cut-off distance for the long-range van der Waals energy term was 10.0 Å. Each system was minimized using the steepest descent algo-rithm up to 50,000 steps, reducing the maximum force to below 1000 kJ/mol/nm. Temperature equilibration was carried out at 300 K using the V-rescale thermostat for 100 ps with a 2 fs time step[86]. This procedure, along with pressure equilibration and the production simulations, was performed in triplicate using different initial velo-cities generated according to the Maxwell–Boltzmann distribution at 300 K. Pressure equilibration was performed with the C-rescale baro-stat at 1 bar for 100 ps, also using a 2 fs time step[87]. Following equili-bration, 20 ns production simulations were performed. The trajectories were sampled every 10 ps for analysis of the production dynamics. The RMSD of alpha carbons was calculated using the 'gmx rms' command in GROMACS. The RMSF of all protein atoms was cal-culated using the 'gmx rmsf' command in GROMACS, focusing on the final 5 ns. The average RMSF values for each atom were mapped onto the protein structures using PyMOL v2.5.4[88].

### Reporting summary
Further information on research design is available in the Nature Portfolio Reporting Summary linked to this article.

## Data availability
The atomic coordinates and cryo-EM density maps have been depos-ited in the PDB and the Electron Microscopy Data Bank (EMDB), respectively, under the following accession codes: 9LDK (AdhE) and 9LDL (BdhE) for PDB. EMD-63003 (AdhE) and EMD-63004 (BdhE) for EMDB. The molecular dynamics result datasets are provided in Zenodo [https://doi.org/10.5281/zenodo.15322847]. The Source Data files for all data presented in graphs within each Figure are also provided as supplementary materials. The previously published PDB entries ana-lysed in this study are available under the following accession codes: 6TQH (*E. coli* AdhE structure in its extended conformation), 6TQM (*E. coli* AdhE structure in its compact conformation), and 6AHC (*E. coli* AdhE structure in its compact conformation). Source data are pro-vided with this paper.

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

## Acknowledgements

We deeply thank members of Dr. Chikara Furusawa's lab and Dr. Wataru Iwasaki's lab for valuable discussions and for helping with computational/experimental setups, especially Mr. Masaki Fujiyoshi, Mr. Kazuki Miyata, Mr. Shun Yamanouchi, Mr. Jun Kuroki, Mr. Keito Watano, and Mr. Yugo Tsunoda. We also appreciate members of Dr. Kozo Tomita's lab and Dr. Misato Otani's lab for sharing devices for protein experiments with kind lectures, especially Dr. Yuka Yashiro, Dr. Seisuke Yamashita, and Ms. Natsu Takayanagi. We are deeply indebted to members of Dr. Masahide Kikkawa's lab for teaching cryo-electron microscopy

experiments, especially Ms. Yoriko Akuzawa and Dr. Chieko Saito. We thank Mr. Tatsuki Tabuchi for helping protein experiments. Finally, we express our sincere appreciation to everyone who gave us feedback on this project, especially Mr. Michihiro Nishimura. This work was supported by the Japan Society for the Promotion of Science (KAKENHI Grant numbers 22J20318 and 24H00870 to N.K., JP22H04925 (PAGS) to Y.N., and 21H05247 and 21H05248 to M.K.), the Japan Science and Technology Agency (GteX JPMJGX23B2 to Y.N. and ERATO JPMJER1902 to C. F.), the ANRI Fellowship to N.K., and by Platform Project for Supporting Drug Discovery and Life Science Research (Basis for Supporting Innovative Drug Discovery and Life Science Research (BINDS)) from Japan Agency for Medical Research and Development (AMED) under Grant Number JP24ama121002 to M.K.

## Author contributions

Conceptualization: N.K., Investigation throughout the study: N.K., Docking simulation and molecular dynamics analysis: S.N., Protein expression: N.K. and K.M., Data visualization: N.K. and S.N., Supervision of cryo-EM analysis: H.Y. and M.K. Methodology: N.K., K.M., S.N., K.O., H.Y., and S.T., Supervision of the study: W.I., M.K., and C.F., Writing—original draft: N.K. and S.N., Writing—review/editing: N.K., K.M., S.N., K.O., H.Y., S.T., Y.N., M.K., W.I., and C.F.

## Competing interests

The authors declare no competing interests.
