## [Peer Review file · Nature Communications]

Repeatability of protein structural evolution following convergent gene fusions

Corresponding Author: Dr Naoki Konno

Version 0:

Reviewer comments:

Reviewer #1

(Remarks to the Author)

The convergent evolution of proteins reveals patterns in genetic adaptation. While local convergence at the residue or domain level is well-documented, global structural convergence through inter-domain interactions is less explored. Here, Konno N. et al. examine the convergent evolution of fusion enzymes BdhE and AdhE, which share similar functions despite evolving independently via different gene fusions. Cryo-electron microscopy showed that BdhE forms donut-like homotetramers, while AdhE forms helical homopolymers. Despite significant structural and amino acid differences, both enzymes share a similar dimeric structure through ALDH-ADH interactions. Interestingly, AdhE oligomers are formed by ADH-ALDH interactions, while BdhE oligomers arise from ADH-ADH interactions. These findings highlight how convergent gene fusions enhance substrate channeling efficiency and provide new insights into molecular evolution.

The section on MD simulation for testing structural stability is intriguing—why not perform simple thermal melt experiments to test them experimentally? One should observe the disintegration of oligomers as the first transition, and comparing these will indicate the strength of intermolecular interaction. It is anticipated that the AdhE oligomer will be less stable than the BdhE oligomer.

The text is well written, although there are some minor errors, such as

- “PDB ID” or “PDB; ID” should be consistent.
- “AlphaFold2” or “AlphaFold v2.3.2.” should be consistent.
- Line 609-614, references should be added in CryoSPARC, UCSF ChimeraX.
- Line 620-625, the unit of box size?. The unit of energy range?

Additionally, the manuscript needs other revisions.

1. Figure 2A, B: Are the loops clearly defined in the cryo-EM map? The maps in Figure S7 seem to lack adequate resolution, especially for AdhE. A model-fitted map with density for these loops is necessary Figure 4 legend, Line 287, The RMSF is typically calculated for individual atoms or residues, not only C . Please revise.
2. Line 289 states that obtaining a p-value of exactly 0 suggests the probability of observing your data (or something more extreme) under the assumption that the null hypothesis is true is so small that it can be considered effectively zero. However, in practice, a p-value of 0 is rarely reported. Instead, researchers typically report very small p-values (e.g., 0.0001 or 1e-10), which indicate extremely strong evidence against the null hypothesis. Please double-check your data.
3. A 5 ns simulation is too short for protein folding or complex stability and may not fully explore the system's conformational space. In Figure S8F, the BdhE tetramer does not appear to have equilibrated, so the results may be influenced by initial conditions. A longer simulation and careful data interpretation are essential needed.
4. Figure 5C, D, S9, S10: As you pointed out earlier, the tertiary structure of BdhE differs from that of AdhE. Since AlphaFold predictions depend on existing databases, the predicted domain structures likely resemble those of AdhE, which may influence the results of homomeric and heteromeric interactions between the split AdhE and BdhE domains.
5. A table of Cryo-EM data collection, refinement, validation, and statistics is required. Additionally, the codes for the cryo-

EM maps and structure deposit are needed.

Reviewer #2

(Remarks to the Author)

The manuscript presents a compelling study on the convergent evolution of protein structures following gene fusions, focusing on the bifunctional enzymes AdhE and BdhE. The work is well-structured, with clear objectives, robust methodologies, and significant findings that advance our understanding of protein structural evolution. The integration of bioinformatics, structural biology, and biochemical assays provides a comprehensive approach to addressing the research questions.

Major comments:

1. While the structural convergence of AdhE and BdhE is well-documented, the manuscript could better address the functional implications of their divergent quaternary structures. For example, how do the helical (AdhE) and ring-shaped (BdhE) structures influence their enzymatic efficiency or regulation *in vivo*?
2. The ecological significance of the non-overlapping phylogenetic distribution of AdhE and BdhE is intriguing but could be explored further. Are there specific environmental factors (e.g., temperature, salinity) that favor one enzyme over the other?
3. The enzymatic assays show that BdhE has lower reaction rates than AdhE. This difference is not fully explained. Is it due to structural differences, substrate specificity, or other factors? Additional experiments or computational simulations could help clarify this point.
4. The molecular dynamics (MD) simulations are relatively short (5 ns). While this is sufficient for initial insights, longer simulations or replica exchange MD could provide a more comprehensive understanding of the structural stability and dynamics of AdhE and BdhE.
5. The manuscript could benefit from a more detailed discussion of how the findings fit into the broader context of protein evolution. For example, how common is structural convergence following gene fusions across other protein families? Are there other examples of enzymes evolving similar functional interfaces through distinct structural mechanisms?
6. Similar traits between AdhE and BdhE (substrate channeling, dimeric form) could emerge from the divergence of the two units before fusion. They may also result from convergent gene fusion. How do the authors classify if similar traits come from divergence pre-fusion or convergence post-fusion?
7. How do the different quaternary structures of AdhE and BdhE contribute to substrate channeling? Can we compare the substrate channeling between the two?
8. Supplementary Figures 3B&C are more informative in concluding the lower reaction rates than Figures 2A&B. Minor correction or comment is needed as the maximum absorbance of NADH mentioned in the main text is 340nm (line 161). However, the assay replicates were measured at 360nm (line 566).

Reviewer #3

(Remarks to the Author)

The manuscript, "Repeatability of protein structural evolution following convergent gene fusions", by Konno and colleagues presents evidence for the convergent origin of ALDH-ADH fusion proteins. The authors present evidence for a previously unknown fusion protein, which they call BdhE, that appears to have evolved, via fusion, independently of the AdhE protein. To support their claims, they present a mix of phylogenetic analysis, experimental structural characterisation, along with some exploratory work that attempts to address whether there is any correlation between presence of AdhE vs BdhE and environment.

My primary expertise is in evolutionary analysis, so I will restrict my comments to this part of the manuscript.

First of all, from the results presented in figure 4A, it does seem there is compelling evidence for the claim that AdhE and BdhE are the result of structural convergence following gene fusion.

In terms of the phylogenetic analyses, these also seem to support the independent origins of these two proteins, but there are some technical issues that need to be addressed. I will lay out my concerns below.

The phylogenies in Figure 1C & D aim to show that AdhE and BdhE have distinct evolutionary origins. To begin on a positive note, this is supported by their non-monophyly, something the authors should note explicitly. Essentially, the point here is that, if there is no way to place AdhE and BdhE on a common branch to the exclusion of other branches, then they are non-monophyletic, which means they do not share a common ancestor. With such a result, there is no need for a rooted tree. I note that the authors present unrooted trees in 1C & D, but present a 'rooted' tree in Figure S2. For the latter, I do not see how they can justify using the *E. coli* AdhE as an outgroup. An outgroup is a distant relative to the 'ingroup' sequences - a simple example might be to root a tree of mammals with sequences from birds, based on prior knowledge that the birds fall outside of the diversity of mammals. There is no clear argument for why *E. coli* AdhE can be used as an outgroup. I would remove mention of this from the main text (l122) and I would remake the tree in Fig. S2. This could be rooted by e.g. mid-point rooting, or (better) left unrooted.

A bigger challenge with the trees in Figure 1 is that the monophyly of each fusion protein (i.e. AdhE and BdhE) might be an artefact. There is insufficient information for me to assess this however. I will first go through the (potential) technical issue with the tree, then I will explain the implication.

From what is presented, it appears that the alignments used as input to build the trees for the ALDH domain (1C) and ADH domain (1D) contain full-length sequences of AdhE and BdhE proteins, plus the respective mono-domain sequences (either ALDH or ADH). From my reading, it appears that this could lead to the AdhE sequences to group together, and the BdhE sequences to group together, because each of these has many more informative sites than the single domain proteins. To be sure that one is tracing the evolutionary relationships between ALDH domains (1C) and ADH domains (1D) I would want to see a phylogeny where only the single domains of each are included. Mixing double and single domain proteins could create an artefact, so it is important to control for this.

Aside from this important test, I note that the authors make a claim for repeated convergence through fusion. However, if the monophyly of each fusion is correctly recovered, then there are only two events, once for emergence of AdhE, and once for emergence of BdhE. In multiple places in the text, the authors suggest fusion has occurred repeatedly (e.g. l45, l322).

If the monophyly of these fusion proteins is an artefact, it actually supports the authors claims that this happens repeatedly. It thus bears closer examination. I specifically recommend repeating the phylogenies to only include the domain from AdhE/BdhE that is relevant to the phylogeny of ALDH or ADH. I should note that, if there are only two events, as the present data seem to show, one should be a little more circumspect with use of terms that imply many more than two convergent events.

The final issue is that the authors note that sequence similarity is low (<30%) across the two fusion enzymes. This calls into question the quality of the underlying sequence alignments. These might be OK, but, as presented, it is not possible to tell - the are only shown in very low resolution. My concern would be that the number of informative sites for phylogenetic analysis is low, which will impact the quality of the phylogeny. I note that, for cases where sequence similarity is very low, it may be challenging to recover a tree, and some researchers advocate switching to using structure for phylogeny (e.g. 38046099, 32302382) instead of sequence. It is not clear from the data if this is a suitable dataset for such approaches - further information is needed in the supplementary material (alignment files).

The alignment in Figure 4 is good in that it shows that loops 3 & 4 and loops 1 & 2 are not conserved across the two fusion proteins. But the alignments aren't in enough detail to see how these two fusion proteins otherwise align. I recommend a supplementary alignment file. Moreover, there are some odd alignments - the AdhE sister ALDH sequences appear to show alignments to the ADH portion (similar but less extreme case seen for the BdhE sister ADH sequences showing some alignment to ALDH domain). I am guessing this is not well aligned and due to varying C- or N-termini - again, alignment files would allow the reader to see this in detail themselves.

Minor matters:

A minor gripe about what the authors have studied vs the reference in the introduction to a 'remarkable number of domain architecture convergence have been suggested by bioinformatic analyses (l176-77). One of those papers (Gough 2005) actually looks at convergence of domain architectures, concluding that this is rare. What the authors are studying is convergence on a multi-domain state via gene fusion. There are many studies that have looked at the rate of fusion/fission of genes, which is much more frequent than domain architecture convergence (Gough notes it is rare). See for example the following articles (PMIDs: 15680510, 16601004).

I found the phrasing in some places to be a little convoluted.

The use of the term >70% unshared amino acids (l43, l96) is awkward. Typically one reports similarity (e.g. <30% identity or 35% similarity).

Please consider whether there is a better way to say 'co-occur/anti-cooccur' (l171).

The Discussion starts with 'Consequently,' (l377) - I would remove this word and begin on the next word.

The statement about the steps in fusion (l405-406) is not well supported. Here the authors are claiming without any substantive evidence that gene neighbouring might be all that is needed for gene fusions. This is really only a correlation, so I would be cautious with a strong claim.

Version 1:

Reviewer comments:

Reviewer #1

(Remarks to the Author)

The authors have addressed all my comments and I support publication of the manuscript.

Reviewer #2

(Remarks to the Author)

The authors improved the manuscript substantially and it is now ready for publication.

Reviewer #3

(Remarks to the Author)

I have read through the revised manuscript and the authors' replies to the issues I raised, and I have examined the new supplementary figure 3 and the three data files. I am satisfied with the responses provided and have no outstanding concerns.

Note that I raised concerns about whether the dataset was too divergent for sequence-based phylogeny, and noted that structural phylogenetic methods provide an alternative for such cases. I think in this case the data look OK - quite a bit of work would be needed to undertake structural phylogenetic analysis, and, looking at the alignments I think this is a case where the alignments are still above the twilight zone, so what the authors have done looks sufficient for the current paper.

REVIEWER COMMENTS

Reviewer #1

Comment 1.0

The convergent evolution of proteins reveals patterns in genetic adaptation. While local convergence at the residue or domain level is well-documented, global structural convergence through inter-domain interactions is less explored. Here, Konno N. et al. examine the convergent evolution of fusion enzymes BdhE and AdhE, which share similar functions despite evolving independently via different gene fusions. Cryo-electron microscopy showed that BdhE forms donut-like homotetramers, while AdhE forms helical homopolymers. Despite significant structural and amino acid differences, both enzymes share a similar dimeric structure through ALDH-ADH interactions. Interestingly, AdhE oligomers are formed by ADH-ALDH interactions, while BdhE oligomers arise from ADH-ADH interactions. These findings highlight how convergent gene fusions enhance substrate channeling efficiency and provide new insights into molecular evolution.

Response 1.0

Thank you so much for your constructive comments. We have revised the manuscript by addressing all the reviewers' comments and improving the overall content. In particular, we thoroughly updated the molecular dynamics results using replicated long-term simulations. We have also improved the explanation and wording of the entire manuscript. Our results suggest that both AdhE and BdhE form helical and circular complexes, respectively, through convergently evolved ALDH-ADH heteromeric interactions and ancestrally inherited ALDH-ALDH and ADH-ADH homomeric interactions. All three types of interactions (i.e., ALDH-ADH, ALDH-ALDH, and ADH-ADH) contributed to the formation of complexes in both families.

Comment 1.1

The section on MD simulation for testing structural stability is intriguing—why not perform simple thermal melt experiments to test them experimentally? One should observe the disintegration of oligomers as the first transition, and comparing these will indicate the strength of intermolecular interaction. It is anticipated that the AdhE oligomer will be less stable than the BdhE oligomer.

Response 1.1

Thank you for your suggestion to conduct the thermal shift assays. Although thermal shift assays can detect thermal denaturation and protein complex dissociation when the hydrophobic regions are exposed during these processes, it is unclear whether subunit dissociation is detectable. Therefore, it is challenging to distinguish between thermal denaturation and subunit dissociation based solely on the fluorescence shifts. Instead, we adopted molecular dynamics simulations because they provide easily interpretable dynamics information, as shown in **Supplementary Video 1**. Based on these movies, what we observed in the molecular dynamics simulations was neither thermal denaturation nor complex dissociation. Rather, the RMSF values indicated subtle fluctuations in the atomic positions that did not disrupt the complex formation. This suggests that thermal shift assays and other experimental characterizations could be useful for investigating the structural dynamics under more extreme conditions (e.g., very high temperatures) than those observed in the simulations presented in this study. We have added a discussion on this point (L471–475).

Comment 1.2

The text is well written, although there are some minor errors, such as

- “PDB ID” or “PDB; ID” should be consistent.
- “AlphaFold2” or “AlphaFold v2.3.2.” should be consistent.
- Line 609-614, references should be added in CryoSPARC, UCSF ChimeraX.

Response 1.2

Thank you for pointing this out to us. We have fixed all these points in our new version.

Comment 1.3

- Line 620-625, the unit of box size?. The unit of energy range?

Response 1.3

Thank you. We have added units for them (L664-666).

Comment 1.4

Figure 2A, B: Are the loops clearly defined in the cryo-EM map? The maps in Figure S7 seem to lack adequate resolution, especially for AdhE. A model-fitted map with density for these loops is necessary.

Response 1.4

Thank you for your comment. We inspected the electron density maps around the loop regions of AdhE and BdhE and confirmed that the density was sufficiently well-defined to resolve the backbone positions (**Supplementary Fig. 9b**). To clarify, the visualized AdhE structure in **Fig. 4a and b** and **Supplementary Fig. 9b** is a previously solved structure of AdhE’s extended conformation structure (PDB ID:6TQH), while the original fig. S7 (**Supplementary Fig. 8** in the revised manuscript) shows the compact conformation of AdhE solved in this study.

Comment 1.5

Figure 4 legend, Line 287, The RMSF is typically calculated for individual atoms or residues, not only C. Please revise.

Response 1.5

We appreciate your comments on molecular dynamics simulations. Based on Reviewer #1’s comment 1.5 and 1.7 and Reviewer #2’s comment 2.4, we re-conducted the molecular dynamics (MD) simulation for 20 ns with three replicates. In this revision, we calculated the RMSF for each atom (**Fig. 4d**). We elaborated on the explanation of this MD experiment in the Results section (L318-337).

Comment 1.6

Line 289 states that obtaining a p-value of exactly 0 suggests the probability of observing your data (or something more extreme) under the assumption that the null hypothesis is true is so small that it can be considered effectively zero. However, in practice, a p-value of 0 is rarely reported. Instead, researchers typically report very small p-values (e.g., 0.0001 or 1e-10), which indicate extremely strong evidence against the null hypothesis. Please double-check your data.

Response 1.6

Thank you. As you suggested, the p-values became zero because they were lower than 2.23e-308, the smallest float number treatable in Python. We have clarified this point in our manuscript (L298).

Comment 1.7

A 5 ns simulation is too short for protein folding or complex stability and may not fully explore the system's conformational space. In Figure S8F, the BdhE tetramer does not appear to have equilibrated, so the results may be influenced by initial conditions. A longer simulation and careful data interpretation are essential needed.

Response 1.7

We appreciate this insightful comment regarding the molecular dynamics (MD) simulations. In response to Reviewer #1's comment 1.5 and 1.7 and Reviewer #2's comment 2.4, we repeated the MD simulations for 20 ns with three independent replicates.

To evaluate whether 20 ns was sufficient to reach the steady state, we calculated the RMSD from the initial structures every 10 ps and examined whether the RMSD changes per 10 ps followed an approximately normal distribution centered around zero. We found that, during the first 5 ns, the distributions deviated significantly from the normal distribution in most simulations. However, for the 15–20 ns interval, the RMSD changes were approximately normally distributed and centered around zero in all cases (**Supplementary Fig. 10b, c**). These results suggest that the systems had reached a steady state by 15 ns and that the latter part of the 20 ns simulations is appropriate for analyzing structural fluctuations.

Owing to changes in the analysis setup, the RMSF results were updated slightly (**Fig. 4d–i**). Notably, **Fig. 4d** now shows a significant difference in RMSF distributions between the AdhE dimer and hexamer and between BdhE dimer and tetramer, while a significant difference was not observed for AdhE in our initial manuscript. This difference is suggested to be caused by the large number of data points in our new version by calculating the RMSF for every atom, not only C-alpha atoms. However, when comparing the atom-wise RMSF differences between the dimeric and multimeric forms, BdhE tetramerization exhibited a greater reduction in RMSF than AdhE hexamerization. Therefore, the tetramer and polymer formation by BdhE and AdhE, respectively, can stabilize the structures compared to forming dimers, and the stabilization effect was suggested to be larger for BdhE, possibly due to the flexibility difference between circular and helical structures. We have elaborated on the explanation in the Results section (L318-337).

Comment 1.8

Figure 5C, D, S9, S10: As you pointed out earlier, the tertiary structure of BdhE differs from that of AdhE. Since AlphaFold predictions depend on existing databases, the predicted domain structures likely resemble those of AdhE, which may influence the results of homomeric and heteromeric interactions between the split AdhE and BdhE domains.

Response 1.8

Thank you for pointing this out to us. As you suggested, for heteromeric interactions, AlphaFold-predicted structures may reflect the reported structures of AdhE, especially for AdhEs and proteins evolutionarily close to AdhE. However, the predicted heteromeric interactions in split BdhE (**Fig. 5d**) were suggested not to be an artifact biased by the previously reported AdhE structures for two reasons: First, the heteromeric interactions were predicted for split BdhE but not for protein families K00132 and K13921, which are evolutionarily closer to AdhE than BdhE (**Fig. 5c**). Second, the heteromeric interaction interfaces of BdhE and AdhE were non-homologous and formed distinct loops (**Fig. 4b**). These results suggest that the heteromeric interaction of split BdhE cannot be predicted simply by information leakage from AdhE structures. We clarified this point in L336-369.

Homomeric interactions are conserved among diverse ALDH families and ADH families (PubMed: 34055881, 30700159), so we agree that the prediction for homomeric interactions in **Supplementary Fig. 11b, c** reflects AdhE and other previously reported structures. However, it did not affect any statements in our study, as we experimentally confirmed the homomeric interactions for AdhE and BdhE (**Fig. 3, 4b**).

Comment 1.9

A table of Cryo-EM data collection, refinement, validation, and statistics is required. Additionally, the codes for the cryo-EM maps and structure deposit are needed.

Response 1.9

Thank you for this suggestion. We created **Supplementary Table 1**, which contains information on cryo-EM data collection, refinement, validation, and statistics, as well as PDB and EMDB accession codes.

Reviewer #2:

Comment 2.0

The manuscript presents a compelling study on the convergent evolution of protein structures following gene fusions, focusing on the bifunctional enzymes AdhE and BdhE. The work is well-structured, with clear objectives, robust methodologies, and significant findings that advance our understanding of protein structural evolution. The integration of bioinformatics, structural biology, and biochemical assays provides a comprehensive approach to addressing the research questions.

Response 2.0

We appreciate your constructive comments. In this revision, we thoroughly updated our manuscript with additional experiments. In particular, we conducted enzymatic assays by varying the salt concentration and found that both AdhE and BdhE exhibited robust enzymatic activities. We also conducted a molecular dynamics simulation for 20 ns with three replicates, and confirmed that our results were robust as explained below.

Comment 2.1

Major comments:

While the structural convergence of AdhE and BdhE is well-documented, the manuscript could better address the functional implications of their divergent quaternary structures. For example, how do the helical (AdhE) and ring-shaped (BdhE) structures influence their enzymatic efficiency or regulation *in vivo*?

Response 2.1

Thank you for pointing this out. The different quaternary structures have different functional implications. Helical AdhE is known to transit between two conformations because of the variability of the helical pitch length (PubMed: 32188856, 32523125). This transition has been suggested to regulate its activity by changing the reachability of substrates to the active site by cofactor binding (PubMed: 32523125). Thus, the helical structure has been suggested to enable the regulation of enzymatic activity *in vivo*. In contrast, BdhE formed a ring structure that would not show such structural freedom as that observed in helical structures (i.e., dynamic pitch length). Thus, the ring structure of BdhE is suggested to confer more rigid enzymatic activity than AdhE robustly against environmental changes (e.g., salinity). Notably, BdhE also showed lower activities than AdhE (**Supplementary Fig. 4b-d**), which may align with the stability-activity trade-off, a general trend across diverse enzymes (PubMed: 37276063, 7831309). We have clarified this point in L455-471.

Comment 2.2

The ecological significance of the non-overlapping phylogenetic distribution of AdhE and BdhE is intriguing but could be explored further. Are there specific environmental factors (e.g., temperature, salinity) that favor one enzyme over the other?

Response 2.2

Thank you for your comments. As BdhE was found to be possessed by species living in ocean and the BdhE we used in our experiment was from a halophilic species (*Halomonas eurihalina*), we hypothesized that salt concentration differently affects the activity of BdhE and AdhE. To verify this, we investigated the enzymatic activities of ethanol oxidization under 2-100 mM NaCl conditions (**Supplementary Fig. 4d**). We found these enzymes showed overall similar activities under all conditions, while AdhE and BdhE showed highest reaction rates at different NaCl conditions (AdhE: 2 mM, BdhE: 100 mM). Therefore, at least, salt concentration does not substantially affect the enzymatic activity difference between AdhE and BdhE tested in this study. We added explanation of the new experiments in our main manuscript (L173-174, 610-613). We would like to test other environmental factors (e.g., temperature and pH) in our future work.

Comment 2.3

The enzymatic assays show that BdhE has lower reaction rates than AdhE. This difference is not fully explained. Is it due to structural differences, substrate specificity, or other factors? Additional experiments or computational simulations could help clarify this point.

Response 2.3

Thank you for pointing out the differences in enzymatic activities. We robustly observed differences in the activities for all four different substrates (**Supplementary Fig. 4b, c**). It is possible that AdhE exhibits higher activity because the current experimental conditions were only partially modified from the conditions for AdhE enzymatic assay in previous studies (PubMed: 31586059). In this revision, we conducted a new experiment to verify the robustness to different salt concentrations, and the tendency was still robust even when the NaCl concentration across 2-100 mM (**Supplementary Fig. 4d**). These results show that the difference in enzymatic activity cannot be explained by differences in substrate specificity and optimal salinity conditions. Because AdhE and BdhE are very different at the amino acid sequence level (<30% sequence identity), there are many potential differences affecting their activities (e.g., active site structures, electric charges, hydrophilicity, and optimal temperatures). The difference in enzyme activity could be due to the complex interactions among these factors, making it challenging to identify the reasons that explain the reaction rate difference. While we focused on functional similarity of AdhE and BdhE to discuss the convergent evolution in this study, we would like to investigate potential factors affecting enzymatic activity difference (e.g., optimal temperature) in our future work.

Comment 2.4

The molecular dynamics (MD) simulations are relatively short (5 ns). While this is sufficient for initial insights, longer simulations or replica exchange MD could provide a more comprehensive understanding of the structural stability and dynamics of AdhE and BdhE.

Response 2.4

We appreciate this insightful comment regarding the molecular dynamics (MD) simulations. In response to Comment 2.4 from Reviewer #2 and Comment 1.5 and 1.7 from Reviewer #1, we repeated the MD simulations for 20 ns with three independent replicates.

To evaluate whether 20 ns was sufficient to reach equilibrium, we calculated RMSD from the initial structures every 10 ps and examined whether the RMSD changes per 10 ps followed a normal distribution centered around zero. We found that, during the first 5 ns, the distributions

deviated significantly from normality in most simulations. However, for the 15–20 ns interval, the RMSD changes were approximately normally distributed and centered around zero in all cases (**Supplementary Fig. 10b, c**). These results suggest that the systems had reached a steady state by 15 ns, and that the latter part of the 20 ns simulations is appropriate for analyzing structural fluctuations.

Owing to changes in the analysis setup, the RMSF results were slightly updated (**Fig. 4d-i**). Notably, **Fig. 4d** now shows a significant difference in RMSF distributions between the AdhE dimer and hexamer and between BdhE dimer and tetramer, while a significant difference was not observed for AdhE in our initial analysis. However, when comparing the atom-wise RMSF differences between dimeric and multimeric forms, BdhE exhibited a greater reduction in RMSF upon multimerization than AdhE did. Therefore, the tetramer and polymer formation by BdhE and AdhE, respectively, can stabilize the structures compared to forming dimers, and the stabilization effect was suggested to be larger for BdhE than for AdhE, possibly due to the flexibility difference between circular and helical structures. We elaborated the explanation in the Result section (L318-337).

Comment 2.5

The manuscript could benefit from a more detailed discussion of how the findings fit into the broader context of protein evolution. For example, how common is structural convergence following gene fusions across other protein families? Are there other examples of enzymes evolving similar functional interfaces through distinct structural mechanisms?

Response 2.5

Thank you for your comments. To the best of our knowledge, this is the first clear report of inter-domain structural convergence following gene fusions; therefore, the generality of this phenomenon is an open question. However, a recent study on citrate synthase showed that homomeric interaction interfaces repeatedly evolved with distinct structural mechanisms, and evolutionary changes were discussed as a free walk without functional changes (PubMed: 39005358). In addition, another recent study found convergent gene fusions of molybdenum insertase domains, and the AlphaFold prediction supported the proximal positions of the two domains (PubMed: 39424966). At least, non-negligible number of convergent domain architecture evolutions have been suggested. Gough (2005) and Forslund et al. (2008) analyzed thousands of sampled domain architectures and suggested convergent evolution for >1% of the analyzed domain architectures (PubMed: 15585523, 18025066). Given the massive number of protein families (>20k entries in Pfam v37.3) and the possible combinations of gene fusions, there are many candidates for convergently evolved fusion proteins, in which we may observe inter-domain interactions converge following repeated gene fusions. We have updated the discussion regarding this point (L436-446).

Comment 2.6

Similar traits between AdhE and BdhE (substrate channeling, dimeric form) could emerge from the divergence of the two units before fusion. They may also result from convergent gene fusion. How do the authors classify if similar traits come from divergence pre-fusion or convergence post-fusion?

Response 2.6

Thank you for pointing this out to us. Our results suggest the post-fusion scenario for two reasons. First, the ALDH-ADH interaction interfaces of AdhE and BdhE were both involved in loop structures that were absent in their closest relative families with a single domain (**Fig. 4b, c**). The most parsimonious evolutionary scenario to explain this observation is that the loop structures were independently inserted into AdhE and BdhE after convergent gene fusions, and interactions evolved following loop acquisition. Second, AlphaFold did not support the heterodimer formation of the closest relative families with a single domain (**Fig. 5c**), also supporting that the ALDH-ADH interaction did not evolve before the gene fusions. Notably, we confirmed that the split AdhE and BdhE as positive controls could be predicted to form dimers, suggesting the validity of the interaction prediction results by AlphaFold (**Fig. 5d**). We have clarified these points in L272-281 and L360-369.

Comment 2.7

How do the different quaternary structures of AdhE and BdhE contribute to substrate channeling? Can we compare the substrate channeling between the two?

Response 2.7

Thank you for your question. As Reviewer #2 expected, the difference in the AdhE and BdhE quaternary structures was suggested to contribute to the difference in the tunnel structures of substrate channeling. Although AdhE forms a substrate tunnel within one dimeric structural unit, the dimeric unit of BdhE has a gap in the wall of the substrate tunnel (L307-314; **Supplementary Fig. 9c-e**). One unique feature of the BdhE tunnel structure is that one dimeric unit can partially fill the gap between the other dimeric units. In addition, structural variations can differ between the AdhE and BdhE substrate tunnels. As helical structures of AdhE enable a change in helical pitch length, the tunnel structure in AdhE can fluctuate (Fig. 4 in PubMed: 32523125), whereas BdhE was suggested not to fluctuate the structure as much as AdhE. We also visualized substrate-channeling structures (**Fig. 3f** and **Supplementary Fig. 9a**). To compare the actual substrate channeling efficiency between the two families, we need to perform additional enzymatic assays, which will be one of our future research directions.

Comment 2.8

Supplementary Figures 3B&C are more informative in concluding the lower reaction rates than Figures 2A&B. Minor correction or comment is needed as the maximum absorbance of NADH mentioned in the main text is 340nm (line 161). However, the assay replicates were measured at 360nm (line 566).

Response 2.8

We appreciate your thorough review of our manuscript. The difference in the wavelengths was because we used different machines (Tecan Infinite 200 Pro M Nano and Tecan Infinite F200 Pro). The measurable wavelengths of the machine for **Supplementary Fig. 4b-d** were limited; therefore, we measured only 360 ± 35 nm absorbance. We have clarified the machine differences in the revised manuscript (L598-600).

Reviewer #3

Comment 3.0

The manuscript, "Repeatability of protein structural evolution following convergent gene fusions", by Konno and colleagues presents evidence for the convergent origin of ALDH-ADH fusion proteins. The authors present evidence for a previously unknown fusion protein, which they call BdhE, that appears to have evolved, via fusion, independently of the AdhE protein. To support their claims, they present a mix of phylogenetic analysis, experimental structural characterisation, along with some exploratory work that attempts to address whether there is any correlation between presence of AdhE vs BdhE and environment.

My primary expertise is in evolutionary analysis, so I will restrict my comments to this part of the manuscript. First of all, from the results presented in figure 4A, it does seem there is compelling evidence for the claim that AdhE and BdhE are the result of structural convergence following gene fusion. In terms of the phylogenetic analyses, these also seem to support the independent origins of these two proteins, but there are some technical issues that need to be addressed. I will lay out my concerns below.

Response 3.0

We sincerely thank you for your comments, especially regarding phylogenetics. In this revision, we have thoroughly checked the multiple sequence alignments used for gene phylogenetic reconstruction. As a result, we confirmed that the non-monophyly was supported by phylogeny estimation from alignments containing only one domain. We have also carefully updated our manuscript, including some wording, based on your comments.

Comment 3.1

The phylogenies in Figure 1C & D aim to show that AdhE and BdhE have distinct evolutionary origins. To begin on a positive note, this is supported by their non-monophyly, something the authors should note explicitly.

Response 3.1

Thank you for your positive comment! We agree that non-monophyly directly supports distinct gene-fusion events. We have explicitly noted this point (L138-140).

Comment 3.2

Essentially, the point here is that, if there is no way to place AdhE and BdhE on a common branch to the exclusion of other branches, then they are non-monophyletic, which means they do not share a common ancestor. With such a result, there is no need for a rooted tree. I note that the authors present unrooted trees in 1C & D, but present a 'rooted' tree in Figure S2. For the latter, I do not see how they can justify using the *E. coli* AdhE as an outgroup. An outgroup is a distant relative to the 'ingroup' sequences - a simple example might be to root a tree of mammals with sequences from birds, based on prior knowledge that the birds fall outside of the diversity of mammals. There is no clear argument for why *E. coli* AdhE can be used as an outgroup. I would remove mention of this from the main text (l122) and I would remake the tree in Fig. S2. This could be rooted by e.g. mid-point rooting, or (better) left unrooted.

Response 3.2

Thank you for pointing this out. We agree that the trees should be left unrooted. We note that the tree visualization in Fig. S2 did not show any roots, although the AdhE sequence was placed at the bottom (i.e., there was trifurcation at the most upstream internal node). Therefore, we have clarified that the trees are unrooted in the revised manuscript (**Supplementary Fig. 2**; L123-124).

Comment 3.3

A bigger challenge with the trees in Figure 1 is that the monophyly of each fusion protein (i.e. AdhE and BdhE) might be an artefact. There is insufficient information for me to assess this however. I will first go through the (potential) technical issue with the tree, then I will explain the implication.

From what is presented, it appears that the alignments used as input to build the trees for the ALDH domain (1C) and ADH domain (1D) contain full-length sequences of AdhE and BdhE proteins, plus the respective mono-domain sequences (either ALDH or ADH). From my reading, it appears that this could lead to the AdhE sequences to group together, and the BdhE sequences to group together, because each of these has many more informative sites than the single domain proteins. To be sure that one is tracing the evolutionary relationships between ALDH domains (1C) and ADH domains (1D) I would want to see a phylogeny where only the single domains of each are included. Mixing double and single domain proteins could create an artefact, so it is important to control for this.

Response 3.3

Thank you for pointing out the possibility that the monophyly of the AdhE and BdhE clades in **Fig. 1c and d** can be an artifact. As the reviewer suggested, this could be true if both ALDH and ADH domains were included in the trimmed alignments for which phylogenies were estimated. However, this possibility is unlikely because we confirmed that the multiple sequence alignments used in Fig. 1c, d only contained the ALDH or ADH domain by trimming the alignments using TrimAl (--gt 0.7). We have added **Supplementary Fig. 3** to show that the number of informative sites used for phylogeny estimation in **Fig. 1** is overall similar between multi-domain and single-domain proteins.

Comment 3.4

Aside from this important test, I note that the authors make a claim for repeated convergence through fusion. However, if the monophyly of each fusion is correctly recovered, then there are only two events, once for emergence of AdhE, and once for emergence of BdhE. In multiple places in the text, the authors suggest fusion has occurred repeatedly (e.g. l45, l322). If the monophyly of these fusion proteins is an artefact, it actually supports the authors claims that this happens repeatedly. It thus bears closer examination. I specifically recommend repeating the phylogenies to only include the domain from AdhE/BdhE that is relevant to the phylogeny of ALDH or ADH. I should note that, if there are only two events, as the present data seem to show, one should be a little more circumspect with use of terms that imply many more than two convergent events.

Response 3.4

We deeply appreciate the suggestion of being circumspect with use of the word “repeated,” because our results indicated each of AdhE and BdhE were monophyletic. In our new

manuscript, we have avoided the word in sentences regarding the evolution of AdhE and BdhE (L100, 102, 279, 338, 377, 455) unless the evolutionary events are explicitly written to occur twice. Note that while the ALDH-ADH enzymes (i.e., fusion enzymes in which ALDH is located upstream of ADH) were suggested to occur twice, ADH-ALDH enzymes (i.e., fusion enzymes in which ADH is located upstream of ALDH) also possibly evolved by gene fusions (L118). Our future research directions include conducting structural similarity searches to find fusion proteins of ALDH and ADH remote homologs and characterizing enzymatic functions and structures to verify whether three or more structural convergences occurred.

Comment 3.5

The final issue is that the authors note that sequence similarity is low (<30%) across the two fusion enzymes. This calls into question the quality of the underlying sequence alignments. These might be OK, but, as presented, it is not possible to tell - they are only shown in very low resolution. My concern would be that the number of informative sites for phylogenetic analysis is low, which will impact the quality of the phylogeny. I note that, for cases where sequence similarity is very low, it may be challenging to recover a tree, and some researchers advocate switching to using structure for phylogeny (e.g. 38046099, 32302382) instead of sequence. It is not clear from the data if this is a suitable dataset for such approaches - further information is needed in the supplementary material (alignment files).

The alignment in Figure 4 is good in that it shows that loops 3 & 4 and loops 1 & 2 are not conserved across the two fusion proteins. But the alignments aren't in enough detail to see how these two fusion proteins otherwise align. I recommend a supplementary alignment file.

Moreover, there are some odd alignments - the AdhE sister ALDH sequences appear to show alignments to the ADH portion (similar but less extreme case seen for the BdhE sister ADH sequences showing some alignment to ALDH domain). I am guessing this is not well aligned and due to varying C- or N-termini - again, alignment files would allow the reader to see this in detail themselves.

Response 3.5

Thank you for your comment. We have attached the alignment files corresponding to **Fig. 4c** as a supplementary dataset (**Supplementary Data 1**). Although the alignment in **Fig. 4c** has an odd part, as the reviewer noted, the alignment itself was not used for phylogeny estimation. We also attached the alignment for phylogenetic estimation corresponding to **Supplementary Fig. 3**. The alignment was prepared by trimming to include only single domains (**Supplementary Data 2 and 3**). The number of informative sites in the datasets was sufficient to support the distinct origins of AdhE and BdhE with >95% UFboot values (**Fig. 1c, d**).

Minor matters:

Comment 3.6

A minor gripe about what the authors have studied vs the reference in the introduction to a 'remarkable number of domain architecture convergence have been suggested by bioinformatic analyses (1176-77). One of those papers (Gough 2005) actually looks at convergence of domain architectures, concluding that this is rare. What the authors are studying is convergence on a multi-domain state via gene fusion. There are many studies that have looked at the rate of fusion/fission of genes, which is much more frequent than domain architecture convergence (Gough notes it is rare). See for example the following articles (PMIDs: 15680510, 16601004).

Response 3.6

We appreciate this comment. As Reviewer #2 also pointed out, we agree that showing quantitative estimates of the convergent evolution of domain architectures, especially through gene fusions, is essential. Although Gough (2005) concluded that domain architecture convergence is rare, the study still identified convergent domain architecture candidates comprising >1% of thousands of analyzed architectures (PubMed: 15585523). Similarly, Forslund et al. (2008) found 12.4% and 5.6% of analyzed convergent architecture candidates in different datasets, with approximately one-third estimated to have emerged independently through domain acquisitions, including fusions (PubMed: 18025066). Furthermore, in the papers highlighted by Reviewer #3, a study investigating gene fusion/fission frequencies across 2,869 domain combinations found that 73% supported only a single gene fusion or fission event, suggesting that multiple fusions and fissions occurred in the remaining domain combinations (PubMed: 15680510). Given that there are >20,000 annotated protein domains and their potential combinations of fused domains, these previous studies support the existence of a substantial number of convergent gene fusion events. We have elaborated on these explanations in the Introduction and Discussion sections (L76-78, L436-446).

Comment 3.7

I found the phrasing in some places to be a little convoluted. The use of the term >70% unshared amino acids (I43, I96) is awkward. Typically one reports similarity (e.g. <30% identity or 35% similarity).

Response 3.7

Thank you. We fixed it (L43-44).

Comment 3.8

Please consider whether there is a better way to say 'co-occur/anti-cooccur' (I171).

Response 3.8

Thank you. We changed the words using "correlated/anti-correlated" (L177, 183, 556).

Comment 3.9

The Discussion starts with 'Consequently,' (I377) - I would remove this word and begin on the next word.

Response 3.9

Thanks! We have deleted this word (L399).

Comment 3.10

The statement about the steps in fusion (I405-406) is not well supported. Here the authors are claiming without any substantive evidence that gene neighbouring might be all that is needed for gene fusions. This is really only a correlation, so I would be cautious with a strong claim.

Response 3.10

We agree with your comment. We deleted the phrase “especially when the gene fusions are facilitated by gene neighboring” because there is no clear evidence to support this facilitation (L428).